# The maternal drug exposure birth cohort (DEBC) in China

Lu Li[1,2,7], Ke Wang[1,2,7], Meixian Wang[1,2,7], Jing Tao[1], Xiaohong Li[1], Zhen Liu[1], Nana Li[1], Xiaoxia Qiu[3], Hongwei Wei[3], Yuan Lin[4], Yuan He[5], Ying Deng[1], Hong Kang[1], Yuting Li[1], Ping Yu ®[1,2] ✉, Yanping Wang[1,2] ✉, Jun Zhu ®[1,2] ✉ & Hanmin Liu[2,6] ✉

Drug exposure during pregnancy lacks global fetal safety data. The maternal drug exposure birth cohort (DEBC) study, a prospective longitudinal investigation, aims to explore the correlation of maternal drug exposure during pregnancy with pregnancy outcomes, and establish a human biospecimen biobank. Here we describe the process of establishing DEBC and show that the drug exposure rate in the first trimester of pregnant women in DEBC (n = 112,986) is 30.70%. Among the drugs used, dydrogesterone and progesterone have the highest exposure rates, which are 11.97% and 10.82%, respectively. The overall incidence of adverse pregnancy outcomes is 13.49%. Dydrogesterone exposure during the first trimester is correlated with higher incidences of stillbirth, preterm birth, low birth weight, and birth defects, along with a lower incidence of miscarriage/abortion. Due to the limitations of this cohort study, causative conclusions cannot be drawn. Further follow-up and in-depth data analysis are planned for future studies.

With the rapid social and economic development together with medical service improvement in China, infectious diseases endangering children's health have been gradually brought under control. However, birth defects and many other adverse pregnancy outcomes remain prominent. Internationally, the 'thalidomide event' in the 1960s was the wake-up alarm for imperative requirement of pregnancy medication safety research[1]. Since then, the pregnancy medication safety has been one of the major, however difficult, subjects in the international maternal-fetal medical research. It was estimated that 25%–99% of pregnant women globally had drug usage[2], and about 80% of them were exposed to drugs of unknown safety information[3]. Due to the special physiological changes in maternal condition and fetal development during pregnancy, exposure to drugs of unknown safety information, including the drugs approved for using by regular

patients, could result in a series of irreversible adverse effects on pregnancy outcomes, such as birth defects, abortion, stillbirth, and so on[4–6]. It was reported that malformations of vital and secondary organs in neonates were associated with drug exposure in pregnant women[7]. However, due to ethical reasons, pregnant women could not participate in regular premarket drug safety tests, and it is virtually impossible to assess the actual drug exposure to fetuses before delivery. In addition, animal safety tests cannot accurately reflect drug teratogenicity in human embryos[8]. Therefore, the effect of maternal drug exposure on pregnancy outcomes needs to be thoroughly studied in larger real-world human populations.

As an important resource platform for etiological research, biobanks play important roles in mechanistic studies and precision medicine research. In recent years, multiple special biobanks

[1]National Center for Birth Defect Monitoring, West China Second University Hospital, Sichuan University, Chengdu, Sichuan, China. [2]Key Laboratory of Birth Defects and Related Diseases of Women and Children (Sichuan University), Ministry of Education, Chengdu, Sichuan, China. [3]The Maternal and Child Health Hospital of Guangxi Zhuang Autonomous Region, Nanning, Guangxi, China. [4]Fujian Provincial Maternity and Children's Hospital, Fuzhou, Fujian, China. [5]National Research Institute for Family Planning, National Human Genetic Resource Center, Beijing, China. [6]Department of Pediatrics, West China Second University Hospital, Sichuan University, Chengdu, Sichuan, China. [7]These authors contributed equally: Lu Li, Ke Wang, Meixian Wang. ✉e-mail: yup@scu.edu.cn; wyxyanping@163.com; zhujun028@163.com; liuhm@scu.edu.cn

have been established from Chinese cohort studies, such as the Taizhou cohort by Fudan University, the Breast Cancer Cohort Study in Chinese Women (BCCS-CW), the Born in Guangzhou Cohort Study[9–11]. The drug exposure and birth cohort (DEBC) in this study has achieved the largest sample size so far in China, with the availability of high-quality and well-annotated human biospecimens besides clinical and epidemiological data, and is at the forefront of Chinese cohort research. The biospecimens of DEBC included maternal peripheral blood, urine, umbilical cord blood, and tissue samples. With these abundant biological samples, our cohort study can provide the opportunities to combine the genetic and environmental factors to study the association between drug exposure during pregnancy and pregnancy outcomes, to explore potential dose-response relationships, and to provide more powerful evidence for mechanistic research.

The DEBC study in China is a prospective, longitudinal, and multicentered cohort study, which ultimately aims to construct a comprehensive database of drug exposure information and associated adverse pregnancy outcomes for Chinese population. The DEBC was launched in August, 2018, and expected to recruit at least 150,000 cases and construct a corresponding biobank. All participants underwent five follow-up visits: in the first trimester (<14 weeks), in the second trimester (22–26 weeks), in the third trimester (32-36 weeks), at delivery, and within 42 days after delivery, respectively. The specific objective was to establish and estimate the correlation between the maternal drug exposure during pregnancy, especially in the first trimester (<14 weeks), and the adverse pregnancy outcomes, such as miscarriage/abortion, stillbirth, preterm birth, low birth weight and significant birth defects.

In this work, a total of 112,986 pregnant women are included in the current analytical database of DEBC, and we show that the drug exposure rate in the first trimester of pregnant women in this database is 30.70%. Among the drugs used, dydrogesterone and progesterone have the highest exposure rates, which are 11.97% and 10.82%, respectively. The overall incidence of adverse pregnancy outcomes is 13.49%. Dydrogesterone exposure in the first trimester shows associations with increased incidence of stillbirth, preterm birth, low birth weight, and birth defects, while displaying a decreased incidence of miscarriage/abortion. There appears to be a lower incidence of birth defects in pregnant women exposed to aspirin during the first trimester compared to those who were not exposed. Causative conclusions cannot be drawn from this cohort study, and the follow-up period of the birth cohort is still limited. This cohort will be further followed up and the data will be analyzed in-depth in future studies.

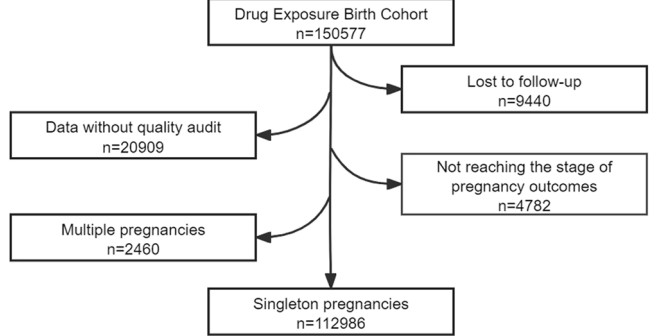

**Fig. 1 | Study design and the number of subjects.** The DEBC has recruited 150,577 participants. After excluding pregnant women lost to follow-up, those without quality audit, reaching the stage of pregnancy outcomes, or with multiple pregnancies, 112,986 pregnant women are included in the current analytical database.

## Results
### Baseline characteristics
Up until 31 December 2021, the DEBC had recruited 150,577 participants. After excluding pregnant women lost to follow-up, without quality audit or reaching the stage of pregnancy outcomes, or with multiple pregnancies, a total of 112,986 pregnant women were included in the current analytical database (Fig. 1). Table 1 shows the baseline characteristics of the study population. Of the recruited pregnant women, 11.42% were of advanced maternal age (≥35 years), 66.74% had a college degree or higher education level, 94.35% had natural conception, and 59.03% had planned pregnancy. Regarding lifestyle factors, 43.23% of pregnant women and/or their husbands

**Table 1 | The demographic characteristics of pregnant women in DEBC**

| Baseline characteristics | N (total = 112,986) | % |
|---|---|---|
| *Maternal age(years)* | | |
| <20 | 1116 | 0.99 |
| 20–24 | 15,308 | 13.55 |
| 25–29 | 44,936 | 39.77 |
| 30–34 | 36,476 | 32.28 |
| ≥35 | 12,904 | 11.42 |
| Missing | 2246 | 1.99 |
| *Maternal education level* | | |
| Primary school and below | 1353 | 1.20 |
| Junior high school | 15,405 | 13.63 |
| High school | 20,554 | 18.19 |
| College | 68,676 | 60.78 |
| Postgraduate and above | 6730 | 5.96 |
| Missing | 268 | 0.24 |
| *Fertilization way* | | |
| Natural conception | 106,597 | 94.35 |
| Assisted reproduction | 5993 | 5.30 |
| Missing | 396 | 0.35 |
| *Planned pregnancy* | | |
| Yes | 66,690 | 59.03 |
| No | 45,938 | 40.66 |
| Missing | 358 | 0.32 |
| *Parental smoking* | | |
| No | 63,207 | 55.94 |
| Yes | 48,844 | 43.23 |
| Missing | 935 | 0.83 |
| *Maternal smoking* | | |
| No | 107,070 | 94.76 |
| Yes | 5053 | 4.47 |
| Missing | 863 | 0.76 |
| *Father smoking* | | |
| No | 64,374 | 56.98 |
| Yes | 47,697 | 42.21 |
| Missing | 915 | 0.81 |
| *Maternal alcohol consumption* | | |
| No | 87,555 | 77.49 |
| Yes | 16,379 | 14.50 |
| Missing | 9052 | 8.01 |
| *Folic acid supplementation* | | |
| No | 11,271 | 9.98 |
| Yes | 100,517 | 88.96 |
| Missing | 1198 | 1.06 |

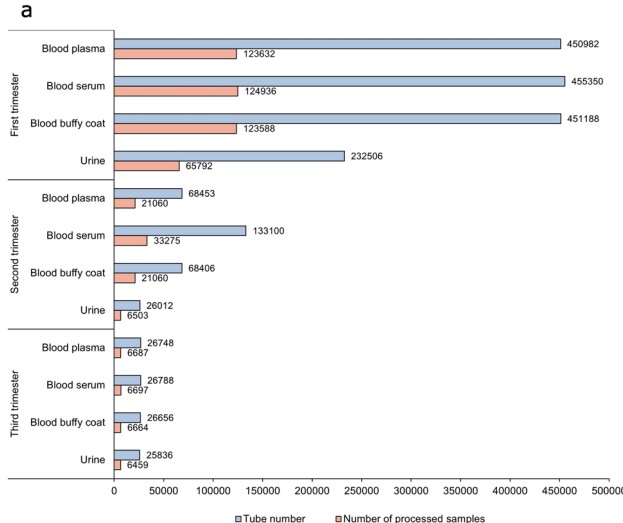

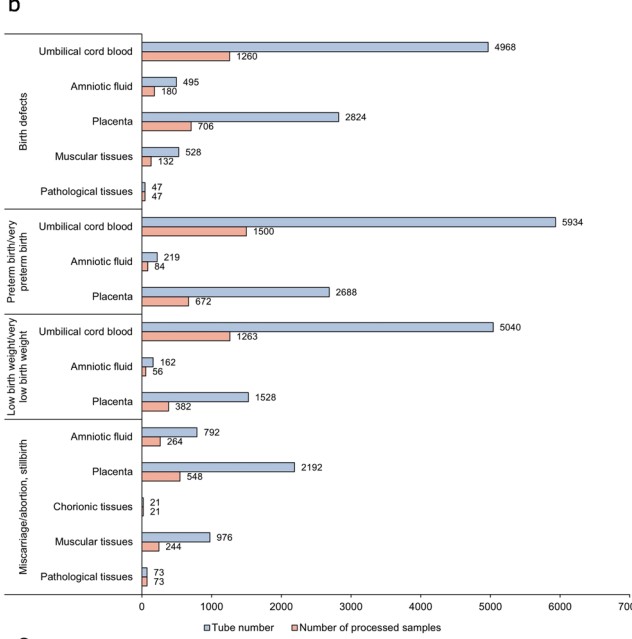

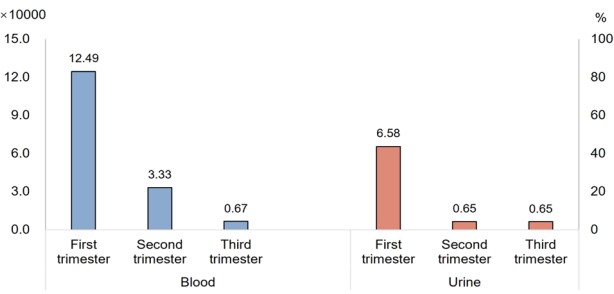

**Fig. 2 | The histogram of biological samples in the biobank. a** The histogram of maternal samples collected during pregnancy. **b** The histogram of biological samples collected from participants with adverse pregnancy outcomes. **c** The number (left, ×10000) and percentage (right, %) of maternal samples collected during pregnancy. Source data are provided as a Source Data file.

Meanwhile, the biobank of this analytical cohort had preserved about 580,000 processed biological samples including about 467,600 processed blood samples (plasma, serum and buffy coat), and about 79,000 urine samples from the pregnant women. Furthermore, ~7400 processed umbilical cord blood samples (plasma, serum and buffy coat), amniotic fluids (supernatants and amniocytes), placenta samples (maternal and fetal surface tissues) and other tissues were collected from the cases with the adverse pregnancy outcomes. About 26,000 processed biological samples from corresponding controls were also stored in the biobank (data not shown). The types and numbers of specimens collected during each pregnancy period and specimens collected for cases with different delivery outcomes are shown in Fig. 2a, b. Among initially recruited 150,577 participants, about 83%, 22%, and 4% of pregnant women provided blood samples in the first trimester, the second trimester, and the third trimester, and about 43%, 4%, and 4% of pregnant women provided their urine samples in the first trimester, the second trimester, and the third trimester, respectively (Fig. 2c). These samples constituted the biobank with rich biological resources for future research.

## Drug exposure

Our survey in 112,986 pregnant women included in this analytical database revealed that over 20 kinds of drugs had high exposure rates during the early pregnancy (Table 2). Among the drugs, dydrogesterone, progesterone, Fuzheng recipe (traditional Chinese medicine), levothyroxine, aspirin, estradiol valerate, allylestrenol, chorionic gonadotrophin, heparin and menotrophin were the top ten frequently used drugs. Dydrogesterone and progesterone had the highest exposure rates, which were 11.97% and 10.82%, respectively. The average duration of medication exposure ranged from 4.1 days (cefaclor) to 58.9 days (progesterone). Of note, there were prominent exposures of Chinese patent medicine including Fuzheng recipe, Ganmaoling granules, Warming and tonifying kidney-yang recipe, and Ban-lan-gen granules (the single largest component is the Isatis indigotica root).

About 30.70% of the pregnant women (34,690) took one or more kinds of drugs in the first trimester. Compared with the pregnant women who did not take any drugs in the first trimester, the pregnant women who took drugs in the first trimester tended to be older, have higher education levels, use assisted reproductive technology, have a planned pregnancy, take folic acid supplementation, and be non-smokers or non-drinkers. The detailed statistical descriptions for these variables are shown in Table 3.

## Adverse pregnancy outcomes

The total incidence of adverse pregnancy outcomes was 13.49%, of which the miscarriage/abortion was 2.52%, the stillbirth 0.77%, the preterm birth 5.15%, the low birth weight 3.87%, and the birth defects 3.56% (Fig. 3). According to the standards of 'Chinese Birth Defects Monitoring Network (CBDMN)', birth defects exclude the micro-malformations, unclassified congenital heart diseases, and atrial septal defects, while these conditions/complications were included in the total incidence of 'adverse pregnancy outcomes' in our study. Maternal age, education level, and fertilization way were all significantly correlated with the occurrence of adverse pregnancy outcomes ($p < 0.05$; Supplementary Table 1). Compared to pregnant women without adverse outcomes, those with adverse pregnancy outcomes tended to be older, have lower education levels (high school or below), and use assisted reproductive technologies (Supplementary Table 1).

## Drug exposure during the first trimester and associated adverse pregnancy outcomes

We assessed the association between medication use during the first trimester and adverse pregnancy outcomes after adjusting maternal age and disease in our database (Table 4). In particular, the incidence of adverse pregnancy outcomes such as stillbirth, preterm birth, low

smoked at least one cigarette daily, 14.50% consumed alcohol at least once during the perinatal period (from 3 months before to 3 months into pregnancy), and 88.96% took folic acid or multivitamins containing folic acid during pregnancy.

**Table 2 | The descriptive analyses of drug exposure rate, formulation, route, dosage, and reasons of drug use during the first trimester**

| Rank | Medicine | Cases | Exposure rate (%) (total n =112986) | Route of administration | Mean exposure days | Average frequency of drug use(%) | | | | Single dose | Reasons of drug use |
|---|---|---|---|---|---|---|---|---|---|---|---|
| | | | | | | QD | BID | TID | <QD | | |
| 1 | Dydrogesterone | 13,528 | 11.97 (11.80, 12.20) | Orally | 26.3 | 9.78 | 56.69 | 32.82 | 0.71 | 10 mg | Threatened miscarriage/low progesterone/assisted reproduction |
| 2 | Progesterone | 12,223 | 10.82 (10.60, 11.0) | Orally | 16.6 | 34.24 | 52.30 | 13.20 | 0.26 | 100/200 mg | Threatened miscarriage/low progesterone/assisted reproduction |
| | | | | Intramuscular injection | 16.3 | 96.72 | 2.38 | 0.25 | 0.65 | 20/40/60 mg | Threatened miscarriage/low progesterone/assisted reproduction |
| | | | | Vaginal administration - capsules | 47.5 | 74.35 | 11.53 | 13.32 | 0.80 | 100/200 mg | Assisted reproduction |
| | | | | Vaginal administration - gel | 58.9 | 96.15 | 3.85 | 0.00 | 0.00 | 90 mg | Assisted reproduction |
| 3 | Fuzheng recipe | 5839 | 5.17 (5.07, 5.33) | Orally | 15.5 | 1.78 | 13.27 | 84.85 | 0.10 | 2g-6g | Threatened abortion |
| 4 | Levothyroxine | 3709 | 3.28 (3.23, 3.33) | Orally | 57.5 | 98.66 | 0.57 | 0.33 | 0.44 | 25/50/75 µg | Hypothyroidism/subclinical hypothyroidism |
| 5 | Aspirin | 3007 | 2.66 (2.61, 2.79) | Orally | 45.5 | 79.52 | 14.02 | 6.06 | 0.40 | 25/50/75/100 mg | Assisted reproduction/anticardiolipin antibody syndrome |
| 6 | Estradiol valerate | 1947 | 1.72 (1.62, 1.78) | Orally | 45.6 | 38.35 | 35.23 | 24.72 | 1.70 | 1/2 mg | Assisted reproduction |
| 7 | Allylestrenol | 1887 | 1.67 (1.62, 1.78) | Orally | 20.9 | 1.80 | 4.79 | 93.35 | 0.06 | 5 mg | Threatened miscarriage/low progesterone |
| 8 | Chorionic gonadotrophin | 1115 | 0.99 (0.94, 1.06) | Intramuscular injection | 5.2 | 86.25 | 0.00 | 0.00 | 13.75 | 1000/2000/5000 IU | Assisted reproduction/threatened abortion/low progesterone |
| 9 | Heparin | 993 | 0.88 (0.84, 0.96) | Subcutaneous injection | 33.1 | 89.37 | 8.50 | 0.34 | 1.79 | 100IU/kg | Assisted reproduction/threatened abortion/high uterine artery |
| 10 | Menotrophin | 764 | 0.68 (0.65, 0.75) | Intramuscular injection | 7.9 | 61.72 | 0.00 | 0.00 | 38.28 | 75IU | Assisted reproduction |
| 11 | Triptorelin | 667 | 0.59 (0.55, 0.65) | Intramuscular injection | 4.3 | 81.25 | 0.00 | 0.00 | 18.75 | 0.1 mg | Assisted reproduction |
| 12 | Amoxicillin | 551 | 0.49 (0.46, 0.54) | Orally | 4.9 | 23.27 | 24.93 | 50.97 | 0.83 | 0.5/1 g | Respiratory tract infection/genitourinary tract infection/periodontitis |
| 13 | Ganmaoling granules | 512 | 0.45 (0.46, 0.54) | Orally | 4.3 | 37.12 | 22.73 | 37.12 | 3.03 | 10 g | Common cold |
| 14 | Prednisone | 487 | 0.43 (0.36, 0.44) | Orally | 44.9 | 92.31 | 3.08 | 4.62 | 0.00 | 5 mg | Assisted reproduction/anticardiolipin antibody syndrome |
| 15 | Urofollitropin | 481 | 0.43 (0.36, 0.44) | Intravenous injection | 9.5 | 62.96 | 0.00 | 0.00 | 37.04 | 75IU | Assisted reproduction |
| 16 | Dimethylbiguanide | 450 | 0.40 (0.36, 0.44) | Orally | 57.8 | 36.65 | 45.55 | 17.80 | 0.00 | 0.25/0.5 g | Hyperglycemia/Assisted Reproduction/Polycystic Ovarian Syndrome |
| 17 | Nifuratel nystatin | 421 | 0.37 (0.36, 0.44) | Vaginal administration | 7.4 | 98.05 | 0.90 | 0.00 | 1.05 | 0.5 g | Vaginitis |
| 18 | Warming and tonifying kidney-yang recipe | 370 | 0.33 (0.27, 0.33) | Orally | 10.3 | 1.69 | 87.57 | 10.73 | 0.01 | 5 g | Threatened abortion |
| 29 | Clotrimazole | 367 | 0.32 (0.27, 0.33) | Vaginal administration | 4.2 | 86.05 | 3.86 | 0.89 | 9.20 | 0.5 g | Vaginitis |
| 20 | Ban-lan-gen granules | 301 | 0.27 (0.27, 0.33) | Orally | 4.8 | 20.44 | 17.52 | 61.31 | 0.73 | 10/15 g | Upper respiratory tract infection |
| 21 | Methylprednisolone | 271 | 0.24 (0.17, 0.23) | Orally | 45.6 | 86.49 | 8.11 | 2.70 | 2.70 | 4 mg | Immune system diseases |
| 22 | Cefaclor | 208 | 0.18 (0.17, 0.23) | Orally | 4.1 | 6.71 | 12.75 | 80.54 | 0.00 | 0.125/0.25 g | Assisted reproduction/upper respiratory tract infection |
| 23 | Cyclosporin | 171 | 0.15 (0.17, 0.23) | Orally | 39.4 | 11.19 | 74.83 | 12.59 | 1.39 | 25/50 mg | Assisted reproduction/threatened abortion/immune system diseases |

QD once a day, BID twice a day, TID three times a day.

**Table 3 | The comparison of demographic characteristics between the drug-exposed and unexposed groups in the first trimester of DEBC**

| Baseline characteristics (total number = 112,986) | Drug exposure in the first trimester | | | | $\chi^2$ | P value |
|---|---|---|---|---|---|---|
| | No | | Yes | | | |
| | N | % | N | % | | |
| *Maternal age* | | | | | 1526.14 | <0.001 |
| <20 | 958 | 1.23 | 158 | 0.45 | | |
| 20–24 | 11,889 | 15.22 | 3419 | 9.81 | | |
| 25–29 | 31,934 | 40.87 | 13002 | 37.31 | | |
| 30–34 | 24,199 | 30.97 | 12277 | 35.23 | | |
| ≥35 | 8009 | 10.25 | 4895 | 14.05 | | |
| Missing | 1147 | 1.47 | 1099 | 3.15 | | |
| *Maternal education level* | | | | | 1145.05 | <0.001 |
| Primary school and below | 1091 | 1.40 | 262 | 0.75 | | |
| Junior high school | 11,830 | 15.14 | 3575 | 10.26 | | |
| High school | 15,085 | 19.31 | 5469 | 15.69 | | |
| College | 46,058 | 58.95 | 22618 | 64.90 | | |
| Postgraduate and above | 4029 | 5.16 | 2701 | 7.75 | | |
| Missing | 43 | 0.06 | 225 | 0.65 | | |
| *Fertilization way* | | | | | 5158.99 | <0.001 |
| Natural conception | 76,284 | 97.63 | 30313 | 86.98 | | |
| Assisted reproduction | 1791 | 2.29 | 4202 | 12.06 | | |
| Missing | 61 | 0.08 | 335 | 0.96 | | |
| *Planned pregnancy* | | | | | 1080.23 | <0.001 |
| Yes | 44,545 | 57.01 | 22145 | 63.54 | | |
| No | 33,554 | 42.94 | 12384 | 35.54 | | |
| Missing | 37 | 0.05 | 321 | 0.92 | | |
| *Parental smoking* | | | | | 565.71 | <0.001 |
| No | 42,526 | 54.43 | 20681 | 59.34 | | |
| Yes | 35,195 | 45.04 | 13649 | 39.16 | | |
| Missing | 415 | 0.53 | 520 | 1.49 | | |
| *Maternal alcohol consumption* | | | | | 3700.13 | <0.001 |
| No | 62,415 | 79.88 | 25140 | 72.14 | | |
| Yes | 12,015 | 15.38 | 4364 | 12.52 | | |
| Missing | 3706 | 4.74 | 5346 | 15.34 | | |
| *Folic acid supplementation* | | | | | 1282.77 | <0.001 |
| No | 9288 | 11.89 | 1984 | 5.70 | | |
| Yes | 68,290 | 87.40 | 32226 | 92.47 | | |
| Missing | 558 | 0.71 | 640 | 1.84 | | |

The *P* values are two-sided.

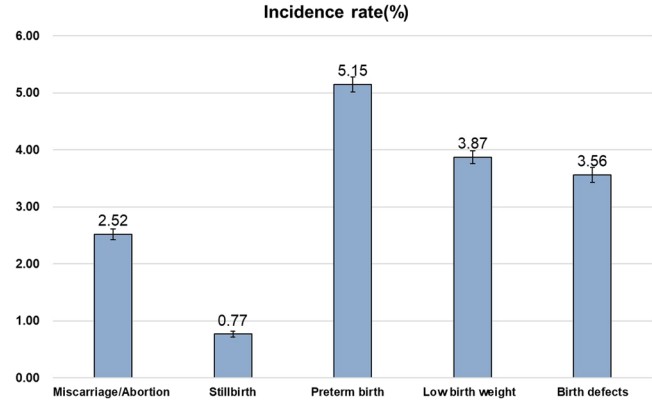

**Fig. 3 | The incidence rate (%) of adverse pregnancy outcomes.** Data are presented as mean values ±standard deviations. The error bars represent 95% confidence intervals (n = 112,986). Source data are provided as a Source Data file.

to validate these findings due to limitations in data collection and the complex composition of proprietary Chinese medicines. There was no significant difference in the incidence of all forms of adverse pregnancy outcomes between levothyroxine-exposed and unexposed group. Infants born to mothers who were exposed to aspirin or amoxicillin during pregnancy had a reduced incidence of birth defects or stillbirth, respectively, compared to infants born to unexposed mothers. Prednisone-exposed group had higher incidence of preterm birth and low birth weight, yet a lower incidence of miscarriage/abortion.

## Discussion

The study of medication usage during pregnancy and adverse pregnancy outcomes is complex and requires large, detailed and realistic databases to capture representative information that could prompt further studies. However, internationally, there have been very rare cohorts specifically focusing on maternal medication exposure during pregnancy. China Teratology Birth Cohort (CTBC), which was expected to recruit at least 300,000 participants[12], was a hospital-based, open-ended prospective cohort study with the aim to assess the risk of birth defects and other adverse pregnancy outcomes associated with maternal environmental and behavioral exposures during pregnancy. The current maternal drug exposure birth cohort (DEBC) was a sub-cohort of CTBC, aiming to explore the impact of maternal drug exposure during pregnancy on pregnancy outcomes, and to establish a biobank of high-quality and well-annotated human biospecimens for further research on the mechanisms of fetal health and diseases. DEBC had recruited 150,577 pregnant women and established a biobank with about 580,000 processed biological samples. This cohort is so far the largest cohort in China that has obtained actual data on pregnancy outcomes and maternal drug exposure during pregnancy. The database collected not only comprehensive information on medication use (including medication start and end dates, accurate doses, accurate gestational ages, etc.), but also a large amount of information on maternal lives, behaviors, environment factors, laboratory test data during pregnancy, and infant health outcome information. Moreover, a large biobank with abundant and diverse biological specimens has been established. It is worth mentioning that this study included traditional Chinese medicine according to the medication habits of the Chinese pregnant population, which would enable us to fill the knowledge gap in evidence-based medicine study on the use of traditional Chinese medicines in Chinese pregnant women. The powerful database overcomes the limited information on medication usage during pregnancy and retrospective bias in previous studies, and is important for exploring the association of medication exposure during pregnancy with pregnancy outcomes. In addition, the establishment of

birth weight, and birth defects, were higher in dydrogesterone-exposed group than those in unexposed group. Meanwhile, there was a decrease in the incidence of miscarriage/abortion in the group exposed to dydrogesterone. There was no observed increase in the incidence of adverse pregnancy outcomes associated with maternal exposure to progesterone during the first trimester. Preliminary findings suggested an association between the use of Fuzheng recipe and increased incidence of most adverse pregnancy outcomes, while the use of Warming and tonifying kidney-yang recipe showed the opposite. The incidence of adverse pregnancy outcomes in the groups exposed to the other two Chinese patent medicines (Ganmaoling granules and Ban-lan-gen granules) showed no significant difference from those in unexposed group. However, further research is needed

**Table 4 | Adjusted relative risks (aRRs) with 95% confidence intervals (CI) from log-binomial multivariate regression model, of adverse pregnancy outcomes associated with maternal medication in the first trimester, adjusted for maternal age and the first trimester maternal diseases that needed treatments (n = 112,986)**

| Drug | Miscarriage/abortion | | Stillbirth | | Preterm birth | | Low birth weight | | Birth defects | |
|---|---|---|---|---|---|---|---|---|---|---|
| | aRR | (95%CI) | aRR | (95%CI) | aRR | (95%CI) | aRR | (95%CI) | aRR | (95%CI) |
| Dydrogesterone | 0.46 | (0.39,0.55)*** | 1.27 | (1.03,1.55)*** | 1.08 | (1.01,1.18)*** | 1.22 | (1.11,1.35)*** | 1.13 | (1.06,1.21)*** |
| Progesterone | 0.90 | (0.77,1.05) | 1.19 | (0.94,1.48) | 1.09 | (0.99,1.19) | 1.09 | (0.98,1.20) | 1.05 | (0.97,1.13) |
| Fuzheng recipe | 1.58 | (1.30,1.90)*** | 1.59 | (1.19,2.10)*** | 0.89 | (0.79,1.01) | 0.85 | (0.73,0.99)*** | 1.36 | (1.22,1.51)*** |
| Levothyroxine | 0.77 | (0.55,1.07) | 0.65 | (0.38,1.13) | 0.93 | (0.75,1.16) | 0.94 | (0.74,1.21) | 1.16 | (0.95,1.40) |
| Aspirin | 0.81 | (0.53,1.19) | 0.55 | (0.29,1.00) | 0.99 | (0.83,1.18) | 1.17 | (0.96,1.43) | 0.83 | (0.71,0.97)*** |
| Estradiol valerate | 1.15 | (0.68,1.87) | 1.39 | (0.70,2.60) | 0.92 | (0.74,1.13) | 0.73 | (0.56,0.94)*** | 0.78 | (0.65,0.94)*** |
| Allylestrenol | 0.41 | (0.25,0.61)*** | 0.54 | (0.28,0.94)*** | 1.21 | (1.01,1.46)*** | 1.20 | (0.95,1.50) | 1.57 | (1.36,1.80)*** |
| Chorionic gonadotrophin | 0.70 | (0.35,1.28) | 0.78 | (0.29,1.78) | 1.09 | (0.85,1.37) | 1.09 | (0.81,1.45) | 1.24 | (1.01,1.52)*** |
| Heparin | 0.81 | (0.39,1.49) | 1.21 | (0.53,2.41) | 1.21 | (0.95,1.51) | 1.09 | (0.82,1.42) | 0.89 | (0.73,1.09) |
| Menotrophin | 0.66 | (0.28,1.41) | 0.92 | (0.24,2.67) | 0.88 | (0.64,1.19) | 0.84 | (0.56,1.22) | 1.19 | (0.90,1.55) |
| Triptorelin | 2.75 | (1.37,5.24)*** | 0.80 | (0.16,2.61) | 1.28 | (0.94,1.70) | 0.85 | (0.56,1.26) | 1.27 | (0.96,1.66) |
| Amoxicillin | 0.96 | (0.58,1.49) | 0.28 | (0.03,0.99)*** | 0.96 | (0.64,1.37) | 1.26 | (0.84,1.81) | 1.18 | (0.81,1.64) |
| Ganmaoling granules | 0.74 | (0.40,1.23) | 0.78 | (0.26,1.80) | 1.07 | (0.71,1.54) | 0.87 | (0.51,1.37) | 1.24 | (0.87,1.70) |
| Prednisone | 0.35 | (0.07,0.99)*** | 0.38 | (0.04,1.41) | 1.37 | (1.01,1.81)*** | 1.51 | (1.07,2.06)*** | 1.03 | (0.78,1.33) |
| Urofollitropin | 1.18 | (0.45,2.72) | 0.22 | (0.01,1.82) | 0.95 | (0.64,1.37) | 1.04 | (0.63,1.62) | 1.18 | (0.86,1.60) |
| Dimethylbiguanide | 2.36 | (1.19,4.21)*** | 1.91 | (0.57,4.72) | 0.82 | (0.54,1.21) | 1.23 | (0.78,1.83) | 1.31 | (0.93,1.78) |
| Nifuratel nystatin | 1.21 | (0.68,2.00) | 1.81 | (0.75,3.85) | 1.05 | (0.66,1.57) | 0.64 | (0.31,1.17) | 1.64 | (1.05,2.43)*** |
| Warming and tonifying kidney-yang recipe | 1.45 | (0.71,2.58) | 0.11 | (0.01,0.97)*** | 1.16 | (0.78,1.64) | 1.12 | (0.70,1.68) | 1.01 | (0.69,1.41) |
| Clotrimazole | 0.65 | (0.26,1.31) | 0.64 | (0.13,1.93) | 1.16 | (0.76,1.70) | 0.96 | (0.53,1.59) | 1.18 | (0.80,1.67) |
| Ban-lan-gen granules | 0.94 | (0.47,1.64) | 0.63 | (0.12,1.88) | 0.78 | (0.41,1.33) | 1.10 | (0.59,1.83) | 0.75 | (0.35,1.37) |
| Methylprednisolone | 0.56 | (0.10,1.79) | 1.95 | (0.56,5.26) | 1.42 | (0.93,2.07) | 1.40 | (0.87,2.16) | 1.18 | (0.81,1.65) |
| Cefaclor | 1.84 | (0.53,5.10) | 0.39 | (0.01,6.19) | 0.92 | (0.51,1.54) | 1.04 | (0.50,1.93) | 0.52 | (0.29,0.86)*** |
| Cyclosporin | 1.79 | (0.45,4.78) | 1.16 | (0.19,4.08) | 1.27 | (0.77,1.98) | 1.37 | (0.77,2.24) | 0.95 | (0.58,1.45) |

Note: Birth defects included uncategorized congenital heart diseases and atrial septal defects diagnosed after birth.
Adjusted relative risks, estimated using log-binomial multivariate regression model adjusting for maternal age and the first-trimester maternal diseases that needed treatments (diabetes, threatened miscarriage, infertility, thyroid disorders, common cold and influenza, vaginitis, thrombosis or antiphospholipid syndrome, hypertension, hepatitis, and other inflammatory diseases).
*CI* confidence interval, *aRR* adjusted relative risks.
The *P* values are two-sided.
***P < 0.05.

a harmonized working mode among different cooperating institutions and a 'green channel' for fast-track diagnosis of suspected birth defects by ultrasound would be a good example to implement and develop other birth cohort projects.

The initial biospecimens were collected from 83% of the recruited 150,577 pregnant women in the DEBC. Because we considered that the first trimester was the relatively most important period for fetal growth and development, of which the mistakes would lead to birth defects, the maternal peripheral blood samples and their processed samples (plasma, serum and buffy coat) during the first trimester accounted for the majority of the biospecimens in the biobank. Fewer biospecimens were collected during the second trimester and the fewest during the third trimester. In the DEBC, 83%, 22%, and 4% of pregnant women provided blood samples in the first trimester, the second trimester, and the third trimester, respectively. Totally, 4% of pregnant women provided their blood samples across the three trimesters of pregnancy. Regarding the urine samples, 43% of pregnant women provided their samples in the first trimester, and 4% in the second and third trimester.

Since this specific study was focused on exploring the association between drug exposure and adverse pregnancy outcomes, the current analytic cohort only included 112,986 pregnant women whose data on maternal drug exposure and pregnancy outcomes were complete and met the quality audit criteria. This is to date the largest cohort with available actual data on maternal drug exposure during pregnancy and

pregnancy outcomes in China, although the present study has not covered most pregnant women across China. Studies covering the entire or most pregnant population in China are not feasible. The factors such as limited financial resources and manpower increased the difficulty of further expanding the sample size for this project. Significantly, this study included collection of a large amount of examination results during pregnancy, collection and preservation of various biospecimens, and multiple time points questionnaire surveys covering a total of 1551 variables. However, based on the information about population demographics, incidence rates of adverse birth outcomes and sample size analysis, the results of this study potentially represented the relatively most accurate assessment of medication use among pregnant women in China. The sample size was calculated according to cohort study requirement during study design, and the actual sample size that we had achieved greatly exceeded the statistical requirement of this study.

Since we included 112,986 pregnant women out of initially recruited 150,577 pregnant women for the current analysis for the reasons further explained in Methods, the representativeness of the current analytical cohort was a concern. Our analysis showed that there were no statistically significant differences in most basic characteristics between the database with 150,577 pregnant women and the current analytical database with 112,986 pregnant women (Supplementary Table 2). The incidence rates of adverse birth outcomes observed in this study were comparable to those reported in other

domestic Chinese investigations and East Asian studies. In particular, the stillbirth rate aligned with East Asian estimates[13], whereas preterm birth and low birth weight incidences were marginally lower than those in Chinese surveillance statistics and research derived from East Asian data[14,15] (Supplementary Fig. 1).

It is very common for pregnant women to use medications due to chronic diseases and in unknown pregnancy situations[16,17]. The proportions of pregnant women taking medications vary by countries or regions. A study using data from two large birth defect studies found that about 90% of women in the United States took at least one medication during pregnancy and 70% took at least one prescription drug[18]. Ninety-three percent of pregnant women received drug prescriptions and dispensations with the usage of an average of 7.4 ± 5.5 different drugs in France, and the 5 most frequently prescribed drugs were paracetamol, iron, folic acid, phloroglucinol and colecalciferol[19]. A survey of the records of the French Health Insurance Service of drug prescriptions during pregnancy in 1000 women living in Haute-Garonne, southwest France, showed that 79% of women were exposed to drugs for which information about safety in pregnancy was not available from animal or human studies[3]. A large sample-size study conducted in Italy showed that 73% of pregnant women in the cohort were prescribed at least one medication during pregnancy[20]. Another registry study showed that about 80% of registered pregnant women in Italy received at least one prescription during their pregnancy, with the most prescribed drugs being folic acid and progesterone[21]. An analysis of the China Health Insurance Association (CHIRA) database in 2015 showed that 11.1% of 7946 sampled pregnant women (in-patients) used at least one medication during pregnancy, and in particular 5.4% of 2896 pregnant women in the first trimester used at least one medication; and the most frequently used medications were intravenous solutions containing vitamins and minerals, progestogens, and antibiotics[22]. Considering that self-medications by pregnant women outside of hospitals were not included in the CHIRA database, there were significant limitations in the data source. In addition, the drug exposure rate might be higher after the implementation of the 'comprehensive two-child' policy since 2016, due to the increasing proportion of pregnant women with advanced maternal age and pregnancy complications. Our cohort study is the first in China to report that drug exposure rate during the first trimester was as high as 30.70% in Chinese pregnant population, with significant differences in the rates of drug exposure among pregnant women with different demographic characteristics. Notably, medications in this cohort did not include nutritional supplements such as folic acid, vitamins and minerals. This study categorized the 'coded drugs' exposure during pregnancy. It was very beneficial for grasping the information about the types of drug exposure to Chinese pregnant women, and would be a treasure trove for evaluation of drug safety in pregnant women.

Dydrogesterone is an orally active progestin, and previous studies indicated that dydrogesterone treatment reduced the risk of abortion[23,24]. A systematic review and a randomized controlled trial concluded that the use of dydrogesterone was effective in the treatment of preterm abortion[25,26]. These results were basically consistent with the finding of the present study. There was not much research on the association between dydrogesterone and adverse pregnancy outcomes. A review in 2009 summarized 28 reported cases of diverse congenital birth defects, with musculoskeletal system defects and complex birth defects being the most common types, followed by masculinization, genitourinary tract defects, neural tube defects, and eye defects[27]. The data did not provide evidence for an association between congenital malformations and dydrogesterone use[27]. In 2015, a case-control study from Gaza reported an association between the drug use and heart defects in offsprings, with an adjusted advantage ratio of 2.71 (95% CI, 1.54-4.24)[28]. However, that study did not adjust for other relevant factors. The present study showed that exposure to dydrogesterone during the first trimester was associated with

increased incidence of stillbirth, preterm birth, low birth weight, and birth defects. As this analysis did not rule out the influence of other maternal environmental and behavioral factors during pregnancy, except for maternal age and diseases, the causal association remains to be further verified.

This study found that maternal exposure to progesterone in the first trimester did not increase the incidence of adverse pregnancy outcomes after adjusting for maternal age and diseases. However, previously reported findings on the association between progesterone exposure during pregnancy and birth defects were inconsistent. Teratogenicity studies in rats[29,30] and monkeys[31] showed that the treatment with progesterone did not cause birth defects. A report from West Jerusalem claimed that progesterone use during pregnancy increased the incidence of birth defects[32], but the study did not control for subjects who had already used progesterone, and mothers of only two abnormal infants were exposed to progesterone during pregnancy. Another project investigated over 500 progesterone-exposed pregnant women, but did not find an association between progesterone exposure and birth defects[33,34]. A collaborative study in West Germany that included 186 progesterone-exposed pregnant women also did not confirm that the hormone increased the risk of birth defects[35]. Other reports from clinical centers did not find an increase in the incidences of birth defects in infants delivered by progesterone-exposed pregnant women[36,37]. A case-control study independently examining hypospadias showed that the use of hormones, including progestin, was associated with an increased risk of the possible abnormalities[38,39], but there was a lack of the relationship between the duration of hormone therapy and the location of hypospadias and its severity. Another study on hypospadias found an odds ratio (OR) of 3.7 (95% CI, 2.3–6.0) in pregnant women with progestin exposure[40], but the information about the doses or the routes of administration was not available from most participants, and the types of progestin were not specified in detail. Our present study did not exclude other maternal environmental and behavioral factors during pregnancy, except for maternal age and diseases that needed medications. We will conduct more in-depth studies in the future. We believe that the large cohort data will add more accurate evidence to future studies on this topic.

The use of proprietary Chinese medicines (pCms) is common in pregnant women in China, and there is no accurate evidence on the effects of these medicines on pregnancy outcomes. The complexity of the composition of pCms has led to few insightful studies on this topic[41,42]. Preliminary analysis of the data from this cohort suggested, for the first time, that the incidence of miscarriage/abortion, stillbirth, and birth defects were increased and the incidence of low birth weight was decreased in the group of pregnant women who took Fuzheng recipe (one of the classical pCms) in the first trimester. The incidence of stillbirth was decreased in the group who took Warming and tonifying kidney-yang recipe in the first trimester. It is worth noting that traditional Chinese medicine supplements are composed of various herbs from different suppliers, and quantitative testing of their active ingredients is still difficult. Furthermore, the dosages of pCms in this study were acquired via questionnaires, hence necessitating more in-depth investigations and additional evidence for substantiation. Despite the limitations of this research, the information reminds us of the urgency and importance of evaluating the safety of these pCms usage during pregnancy.

Thyroid hormones are crucial for fetal development. The frequency of pregnancy with overt hypothyroidism was reported to be 0.3%–0.5%[43]. The most commonly used thyroxine medication during pregnancy is levothyroxine in China. In this study, we found that the exposure rates of levothyroxine in the first trimester was 3.28% in DEBC, and no statistically significant difference was found in adverse pregnancy outcomes between levothyroxine-exposed and unexposed groups. An observational cohort study showed that levothyroxine had

no effect on the rate of live births in patients with recurrent miscarriage associated with subclinical hypothyroidism (SCH)[44]. The use of levothyroxine did not result in a higher rate of live births in normal thyroid-functioning women with thyroid peroxidase antibodies[45,46]. These results are in line with the findings of this study. In contrast, data from 13 cohort studies and randomized controlled trials showed that pregnant women with SCH treated with levothyroxine had a lower rate of pregnancy losses (OR 0.78, 95%CI 0.66–0.94)[47]. Dhillon-Smith et al. concluded that the use of levothyroxine in women with SCH is a key therapeutic measure to prevent miscarriage based on the latest available evidence[48]. Therefore, further research is needed to evaluate the efficacy of levothyroxine treatment in pregnant women with SCH and its impact on pregnancy outcomes.

Additionally, the study indicated that the use of aspirin during the first trimester was associated with a decreased incidence of birth defects. The finding of the relatively high rate of exposure to aspirin and heparin in pregnant women in China is particularly interesting. Thrombophilia has been implicated in adverse obstetrical events such as miscarriage, recurrent miscarriage, intrauterine growth restriction, severe pre-eclampsia, and placental abruption. There was also reasonable evidence to suggest that some cases of miscarriage and recurrent miscarriage were associated with thrombosis of placental vessels and infarction[49]. Heparin can exert anti-clotting effects by increasing the action of antithrombin, the natural anticoagulant. According to Chinese experts consensus on prevention and treatment of spontaneous abortion with low molecular weight heparin (LMWH) (2018) and Chinese expert consensus on diagnosis and management of recurrent spontaneous abortion (2022), LMWH and low-dose aspirin (LDA) are recommended in China to reduce the risk of adverse pregnancy outcomes such as miscarriage, recurrent miscarriage and venous thromboembolism[50,51]. Therefore, the evaluation of the safety of these drugs during pregnancy is urgent. In the future, we intend to conduct specialized analyses on the safety of various drugs exposed during pregnancy.

The study also assessed the maternal exposure to other drugs. The incidence of stillbirth was decreased in pregnant women exposed to amoxicillin during the first trimester with an aRR of 0.28 (95% CI, 0.03-0.99), compared to that in unexposed group in the present study. Amoxicillin is widely used to treat infectious diseases during pregnancy. There is no direct evidence in the literature that amoxicillin taken during pregnancy improves stillbirths. However, infections during pregnancy are an important cause of stillbirth, and more than 40 bacterial, viral and other pathogens have been associated with stillbirth[52]. About 50% of stillbirths or more may be caused by infection in low- and middle-income countries[53]. Therefore, we speculate that taking amoxicillin during pregnancy may improve maternal infection status and have a protective effect against stillbirth.

According to a randomized controlled trial, exposure to prednisone seemed to be an independent factor contributing to the risk of preterm birth, because pregnant women treated with prednisone in combination with low-dose aspirin were more likely to have preterm labor than patients treated with aspirin alone[54]. Another multicenter randomized trial also found higher frequencies of 'serious' maternal morbidity and preterm delivery among women randomly assigned to prednisone group than those in the group using low-dose heparin[55]. Prednisone is a corticosteroid, and some studies have shown that the use of higher doses of corticosteroid was associated with an increase in incidence of preterm birth[56,57]. These findings are similar to our results. Yet, deeper studies on the independent effects of prednisone are required.

Furthermore, it is notable that we need to balance the risks of using medication against potential consequences of not using them. For example, untreated pregnant women with preeclampsia have a greatly increased risk of seizures, endangering the health of both themselves and the fetuses[58]. On the contrary, cautious use of anti-epileptic medication can reduce the risk[59,60]. Similarly, our findings indicated that the use of dydrogesterone was linked to a lower incidence of miscarriage or abortion. Careful consideration of risks and benefits, accounting for gestational timing and dosage, is warranted before taking medication. However, there is still limited research on the differences in maternal and fetal outcomes associated with different medication regimens (different medications, doses and durations) for the same disease, as well as in-depth research combining internal and external exposures. These aspects of research will be addressed in future studies based on this cohort.

The limitations of this study were the following: (i) Causative conclusions cannot be drawn from this cohort study, and the follow-up period of the birth cohort is still limited. Further follow-up is needed for the discovery of association of metabolic and functional birth defects with drug exposure during pregnancy and for assessment of the long-term correlation of early life medication exposure with human health. (ii) The present preliminary analysis did not rule out the influence of other maternal environmental and behavioral factors during pregnancy, except for maternal age and diseases that needed medications, thus the results of the current analysis should be viewed with caution. (iii) Based on our original research design, no blank samples (tubes) were stored in our biobank to check for sample contamination. Yet, because the collection and processing of samples were carried out in clinical laboratories, in which the current national clinical laboratory standards and regulations were strictly followed, we judged that the possibility of sample contamination was very small but could not be absolutely excluded. (iv) The findings of this study were based on reports of drug exposure during the first trimester of pregnancy. Some of the pregnancy outcomes might result from drug exposure later in pregnancy, requiring further analysis of medication use throughout gestation to precisely correlate drug exposures with pregnant outcomes. (v) The drug dosages in this study were obtained from questionnaires, lacking internal exposure data. Future research will integrate both internal and external drug exposure information to provide stronger evidence for the safety assessment of medication during pregnancy. (vi)While the current sample size satisfied this study's requirement, the cohort of 150,577 was still relatively small compared to China's approximately 10 million annual deliveries. Since comprehensive national data on the characteristics of pregnant women are unavailable, it remains uncertain whether the present cohort accurately represents the overall pregnant population in China. However, the present study might be the most rigorous investigation to date on medication use among pregnant women in China.

Future studies should focus on stratified analysis based on specific maternal diseases and investigation of the connections between medications during pregnancy and adverse pregnancy outcomes, accounting for exposures timings, durations, routes, and doses.

## Methods

### Ethics

The DEBC was approved by the Ethics Committee of Sichuan University, China (Protocol ID: K2017045), and followed the tenets of the Declaration of Helsinki. 'Written informed consent" from the patients were collected by the collaborating hospitals of DEBC when the patients were admitted to the hospitals. This study has obtained all relevant approvals from the Ministry of Science and Technology of China for this work.

### Cohort recruitment

The cohort study was launched on August 1, 2018, and aimed to recruit 150,000 pregnant women. The recruitment was implemented with the cooperation of 49 hospitals at different levels (including tertiary or secondary hospitals) from 15 Provinces in China, of which Sichuan Province made the major contribution[12].

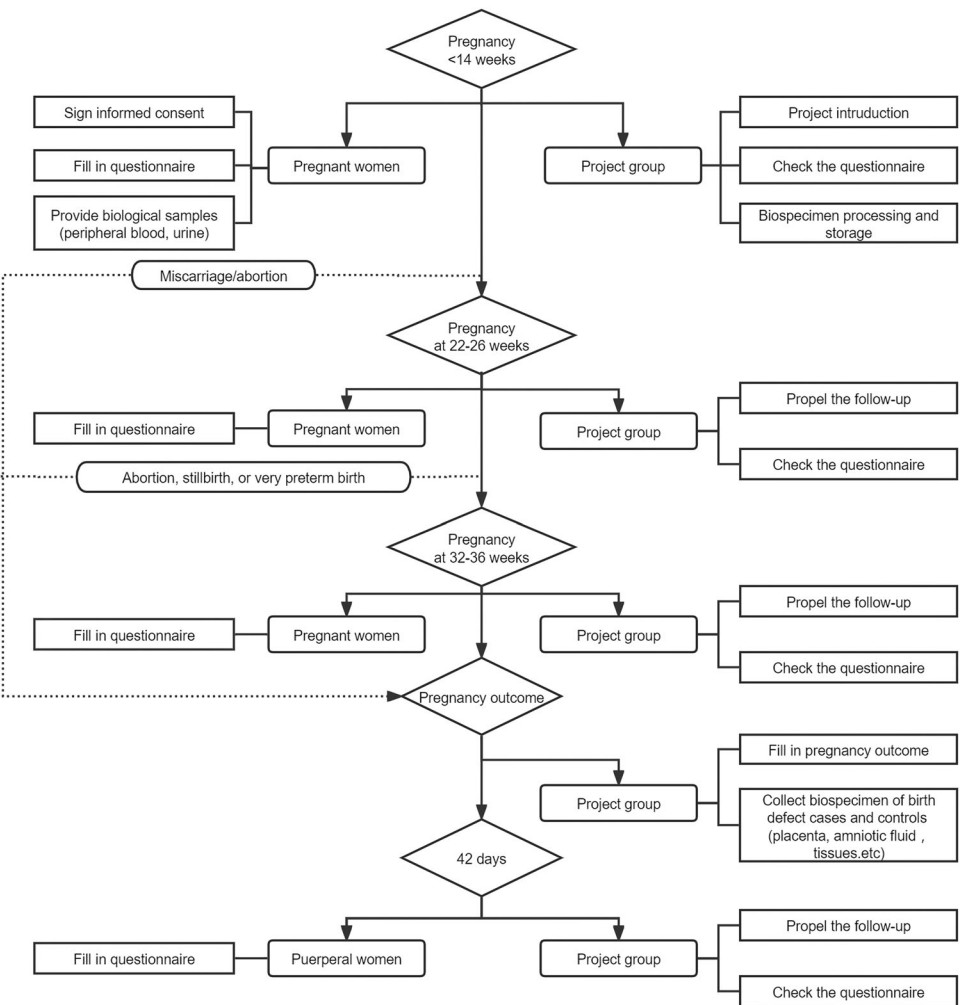

**Fig. 4 | Flow chart of measurements and time points in the cohort.** All participants undergo five follow-up visits: in the first trimester (<14 weeks), in the second trimester (22–26 weeks), in the third trimester (32–36 weeks), at delivery, and within 42 days after delivery.

Subjects eligible for cohort recruitment must meet the following criteria: Pregnant women who (i) had gestational age less than 14 weeks upon first prenatal visit; (ii) planned to establish their health records and deliver in collaborating hospitals; and (iii) were well informed about this study and willing to cooperate with the research and sign the informed consent. Pregnant women were excluded from the study if: (i) they had mental health diseases and could not cooperate with questionnaire investigation; (ii) the fetus had been dead in utero at the time of the first prenatal visit; (iii) they were not willing to cooperate with the research; and (iv) they were HIV-positive.

The 49 collaborating hospitals care for the general population of pregnant women, rather than a selected sub-population. Approximately 190,000 pregnant women come for antenatal care per year, of which 95% receive antenatal care throughout their entire pregnancy and delivery period. Among those 95% of pregnant women, around 30% met the inclusion criteria for this study. Ultimately, 177,150 pregnant women were approached, of which 150,577 consented to DEBC data collection, and a biobank with about 580, 000 processed biological samples has been established.

**Follow-up strategies**

In DEBC, pregnant women (participants) were recruited in the first trimester (usually between 6 and 14 weeks of gestation). At this point, each recruited participant was asked to complete questionnaires, and donate 8 mL peripheral blood (4 mL with EDTA and 4 mL without EDTA), and 20 mL urine (Fig. 4, and Supplementary Table 3). About 17%

of recruited participants did not donate any sample. The recruited participants were followed up in the second trimester (22-26 weeks), the third trimester (32-36 weeks), at delivery, and within 42 days after birth (Fig. 4). The designed structured questionnaires were given at each pregnancy stage through a mobile APP or face-to-face interviews[12]. Maternal medical charts, clinical laboratory measurements from each follow-up and the pregnancy outcomes were abstracted from medical records by our obstetricians or investigators. When the case with fetus of a birth defect was diagnosed, the investigator was asked to stamp a star on the mother's medical records for the follow-ups. For a suspected birth defect, we established a 'green channel' for fast-track diagnosis, meaning that when the fetus was suspected of a birth defect, the doctor could apply for a 'green channel' for the mother for consultation with three skilled and experienced sonologist in the West China Second University Hospital, Sichuan University.

The birth cohort study also aimed to establish a biobank containing diverse biospecimens. The standard operating procedures (SOP) were followed when collecting, processing and storing biospecimens such as samples of peripheral blood, urine from pregnant women, and amniotic fluid, umbilical cord blood, tissues from cases of adverse pregnancy outcomes, and controls in all collaborating medical institutions (Fig. 5a and Supplementary Table 4). According to the SOP, plasma, serum, and buffy coat were prepared from each blood sample; amniocytes and supernatant were prepared from each amniotic fluid sample. All blood samples were aliquoted within 2 h and then frozen at

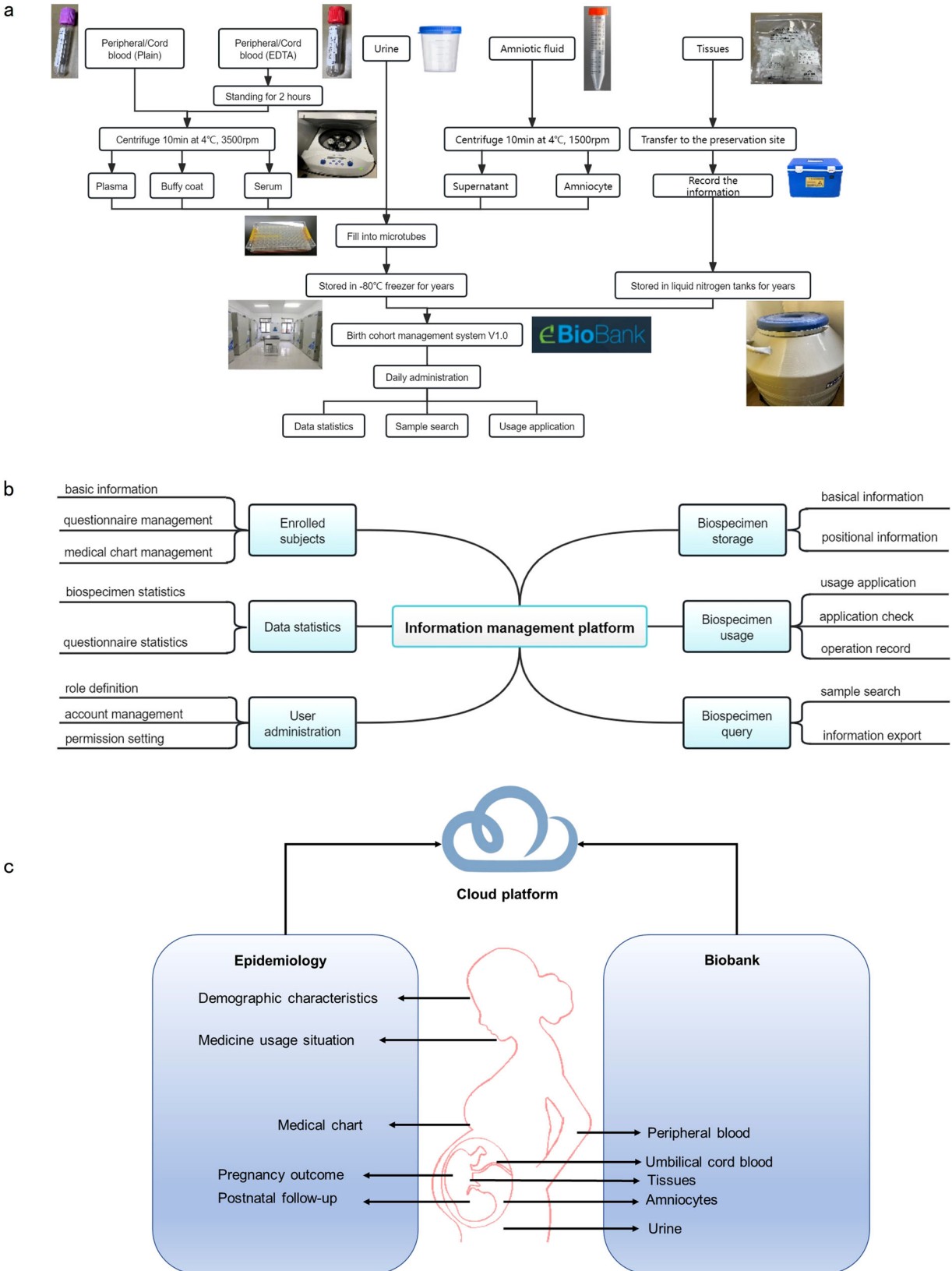

**Fig. 5 | The diagrams of epidemiology and biobank information in DEBC. a** Flow chart of the biospecimen processing, storage, and management. **b** The presentation of the information platform. **c** The main content in the DEBC study.

−80 °C, and the dissected tissues were immediately aliquoted and frozen at −80 °C.

When pregnant women had miscarriage/abortion, stillbirth, or very preterm delivery (<32 weeks), they would skip the rest of follow-ups, and the obstetricians or investigators would complete the pregnancy outcome records directly (Fig. 4). When adverse pregnancy outcomes occurred, the related biological samples were collected by experienced doctors or nurses, according to SOP (Supplementary Tables 3 and 4). For every case of adverse pregnancy outcome, one control (normal fetus whose mother or parents signed the informed consent) was identified in the same research institution.

## Data collection

The DEBC collected a large amount of data, thoroughly covering the maternal, paternal and fetal conditions. The first structured questionnaire given in the first trimester included eight parts, covering the basic information about parents, pregnancy history, health status, working status, living environment, health behavior, diet and nutrition, and sleep condition. The health status part of the questionnaire focused on the detailed information about drug exposure, including drug name, exposure time and duration, route, dose, frequency, and reason of administration during pregnancy. The detailed information in this questionnaire has been reported[12]. The questionnaires given in the second and the third trimester mainly focused on the medication, pregnancy complications, nutraceuticals, and sleep condition. The investigators verified the drug information (including prescription drugs and over-the-counter drugs) in hospital medical records, and made supplements or modifications in the questionnaires (if necessary) to ensure that drug exposure information was collected comprehensively and accurately. The maternal clinical data, including fundal height, abdominal girth, blood pressure, ultrasonic inspection results and clinical laboratory results, were also abstracted from medical records during the pregnancy.

The pregnancy outcomes including delivery/termination way of pregnancy, gestational age, detailed discharge diagnosis, infant sex, birth weight, length, head circumference, birth defect status and neonatal disease screening results, were abstracted from medical records by trained investigators. The questionnaire given within 42 days after birth was about neonatal health and possible newly arisen neonatal diseases.

Biological samples were collected under different scenarios. All recruited pregnant women were asked to donate peripheral blood and urine samples on the first investigation. Chorionic tissues or placenta samples were collected when the miscarriage/abortion or stillbirth occurred with unknown reasons in pregnancy. When a fetus was diagnosed with a birth defect, amniocytes were collected through amniocentesis during the second trimester. In the case of an abortion or a birth defect, amniotic fluid, placenta tissues, umbilical cord blood and fetal muscular tissues, pathological tissues were collected. Umbilical cord blood and placenta tissues were collected from women with very preterm birth, with very low birth weight neonates (<1500 g), and from controls with normal fetuses (Supplementary Table 3).

## Quality control

The recruitments, follow-ups and biospecimen collections were managed by an information platform (birth cohort management system V1.0). This system combined the collected questionnaire data, medical examination results, clinical laboratory results and biospecimen information (Fig. 5b, c). Questionnaire data could be collected through mobile phones, personal computers, or manual inputs. The platform used various automated and intelligent measures to ensure high-quality data entry, including the logic limit and intelligent reminder in data entry, traceability for data collection process, and real-time feedback and error correction. The compliance of participants in follow-ups was improved through timely sending of messages and reminders. There were specialized quality management teams responsible for the maintenance of database and the biospecimen bank, respectively. They would check the data entries regularly and contact the responsible investigators to fill in omissions and correct errors. During the project implementation period, the researchers and staff with access to the research data signed the confidentiality agreement to ensure the data security. Meanwhile, every management account had its own permissions for the access to the questionnaires, biospecimens, or other responsible data. The collected data on the platform were uploaded daily and stored in the designated server in West China Second University Hospital, Sichuan University. The platform had sufficient security measures in the network front-end, data transmission and data storage.

We also implemented strict quality control measures for our questionnaire data. Surveyors for questionnaires received standardized training. The questionnaires were filled by recruited pregnant women under the guidance of a surveyor during their one-on-one meetings. After uploading the questionnaire, the system would check for missing data and logical errors. Additionally, the contents of the questionnaires were reviewed daily to ensure that the information collected was complete. A time period was set for each interview node, requiring interview completion to reduce recall bias. For all maternal and fetal health outcomes, there was a special quality control team to assess the quality of the medical health records from each sub-center to ensure the data accuracy. All laboratory tests were conducted locally at each collaborating hospital with regular quality control measures. In addition, we established ultrasound examination cooperation channels among our partners, and all suspected birth defect cases that could not be diagnosed locally could be sent to the hospital with the strongest diagnostic capability for diagnosis to ensure the accuracy of diagnosis of all birth defect cases.

The participants excluded from this analytic cohort included the following: (i) the cases that were lost to follow-up, meaning that the four follow-up questionnaires including those in the first trimester, the second trimester, the third trimester, and pregnancy outcomes were incomplete; (ii) the participants whose data failed to pass quality audit, meaning that the important data such as medication, pregnancy complications, neonatal delivery, etc. had not yet been verified in the hospital information system; (iii) the cases that had not reached the stage of pregnancy outcomes, meaning that the cases had not yet reached the final follow-up cut-off dates for pregnancy outcomes, such as pregnant women who were still in the second or the third trimester, or had not yet delivered; (iv) the cases of multiple pregnancies, including twins and triplets. Pregnant women with multiple pregnancies were excluded from this analytic cohort because the unit of the analysis was one fetus, and including multiple pregnancies would duplicate the calculation of a pregnant woman's exposure to a certain drug during pregnancy, which might introduce bias and affect the validity of the results.

Besides, we regularly export and analyze the data to identify potential outliers. Considering that most quantitative data collected in this study followed a normal distribution, outliers were defined based on the 3-σ principle (if the absolute residual error of a measurement exceeded 3σ, it was considered an outlier). For identified outliers, we first cross-checked against the hospital information system and national monitoring systems. For data that could not be verified, telephone follow-ups were conducted for confirmation and correction. For the small portion of outliers that simply could not be rectified through the above steps, given our large sample size and the rarity of such outliers, we treated them as missing data.

The biospecimens collected for this study were also subjected to strict quality control measures: (i) The operators in each sub-center received standardized training; (ii) Each sub-center followed the same set of SOP for biospecimen collection; (iii) All biospecimens were collected in ice boxes and processed in sub-centers within 2 h after

collection. Then each type of processed biospecimen from the same participant was processed into different vials which were immediately stored in −80 °C refrigerators. The frozen biospecimens were placed in dry-ice boxes (temperature ranging from −70 °C to −78 °C) which were transported across cities; (iv) These dry-ice boxes and refrigerators were equipped with internal temperature control and monitoring systems to ensure the absolute safe storage temperature; (v) All processed peripheral blood samples and urine samples were finally stored in −80 °C refrigerators, and biospecimens from placenta, muscular tissues and pathological tissues were stored in liquid nitrogen tanks at temperatures below −150°C in West China Second University Hospital. For final storage, multiple vials containing each type of processed biospecimens such as plasma, serum, and buffy coat from the same pregnant woman were not placed in the same box or refrigerator in principle, but were stored in different boxes and refrigerators to prevent unexpected problems caused by possible sample losses or breakdown of one refrigerator.

### Data standardization

The drugs taken by pregnant women in this study were coded individually using the Social Security Drug Classification and Codes (LD/T90-2012), by which the nutritional supplements such as folic acid, vitamins, calcium, and iron were not included. Thus these nutritional supplements were not included in the drug exposures in the present cohort study. Adverse pregnancy outcomes included five conditions: miscarriage/abortion, stillbirth, preterm birth, low birth weight, and birth defects. Stillbirth was defined as fetal death in utero during delivery after 22 weeks of gestation; preterm birth was defined as delivery before 37 weeks of gestation in live birth; low birth weight was defined as birth weight <2500 g in live birth; and birth defects were defined with reference to the Chinese Birth Defects Monitoring Network (CBDMN) and International Clearinghouse for Birth Defects Surveillance and Research (Supplementary Table 5 and Supplementary reference). All birth defects were coded using ICD-10 for standardization (Supplementary Table 6).

### Statistical methods

This study used Pearson's $\chi^2$ test to compare the differences in demographic characteristics between the drug-exposed and unexposed groups in the first trimester. We used log-binomial multivariate regression to estimate adjusted relative risks (aRRs) for the association of maternal medication in the first trimester with adverse pregnancy outcomes after adjusting maternal age and the first trimester maternal diseases that needed treatments (diabetes, threatened miscarriage, infertility, thyroid disorders, common cold and influenza, vaginitis, thrombosis or antiphospholipid syndrome, hypertension, hepatitis, and other inflammatory diseases). $P < 0.05$ was considered statistically significant. All statistical analyses were conducted using STATA® (Version 12.1, StataCorp LLC, College Station, TX, USA) and SAS® software (Version 9.4, SAS Institute Inc., Cary, NC, USA).

### Reporting summary

Further information on research design is available in the Nature Portfolio Reporting Summary linked to this article.

## Data availability

The raw data is not publicly available due to data privacy laws, according to the China's Ministry of Science and Technology. Individual data should be requested by email to the corresponding author (email: zhujun028@163.com) and must include the name and full contact information of the person and institution requesting the data, the research objectives, methodology, anticipated outcomes, plans for sharing results, and the purpose of requesting the data. Data requests under agreement will be subjected to appropriate confidentiality obligations and restrictions. Please expect a processing timeframe of approximately 6–8 months for data requests. Source data are provided with this paper.

## Code availability

The statistical models' code is available in Supplementary Software 1. This includes SAS code files for log-binomial multivariate regression test, along with an introduction file of variables. The analytic SAS code is available through a git repository at https://github.com/wangkehope0839/log-binomial-multivariate-regression.

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

## Acknowledgements

The work was supported by the National Key Research and Development Program of China (NO.2022YFC2704701, 2016YFC1000102), the scientific research project of the Sichuan Province (NO.2019YFS0530), and Sichuan Science and Technology Program (No. 2022NSFSC0656, 2022NSFSC0661). We are grateful to all collaborating medical institutions and staff, especially the West China Second University Hospital, Sichuan University, for their substantial support that enabled the study to be implemented smoothly and effectively.

## Author contributions

L.L., K.W., and M.W. are joint first authors. P.Y., Y.W., J.Z., and H.L. designed the study and analyzed the data. L.L., K.W., and M.W. drafted the paper. J. T., X.L., Z.L., N.L, X.Q., H.W., Y. L. (Yuan Lin), Y. H., Y.D., H.K., and Y.L. (Yuting Li) contributed to the critical interpretation of the results, data quality control, and drug coding. P.Y., Y.W., J.Z., and H.L. proofread the manuscript and contributed equally. All authors have seen and approved the final version. All authors have read and confirmed that they meet ICMJE criteria for authorship.

## Competing interests

The authors declare no competing interests.
