## [Peer Review File · Nature Communications]

REVIEWER COMMENTS

Reviewer #1 (Remarks to the Author):

The submission “The maternal drug exposure birth cohort (DEBC) in China” describes the potential adverse effects of drug exposure on pregnancy complications and birth outcomes based on 112,986 pregnant women from The maternal drug exposure birth cohort (DEBC) in China. Clarifying the association of drug use with healthy outcomes is a great and important work for China, but this paper appears to have not been well conducted. Several main concerns should be taken into considerations.

1 The pregnant women usually intake drug because of some medical condition, resulting in a huge bias when researching the associations between drug and adverse outcomes. The medical condition, but not the drug, might be the reason of the pregnant complications and adverse birth outcomes. The drug usage might actually reduce the risks of adverse outcomes when treated the medical conditions. It is of great importance to clarify this bias. The authors should be more rigorous when draw conclusion. Additionally, the cohort collected he drug exposure time, duration, route, dose and reason in the questionnaire. These are key information but it seems that the authors did not take fully consideration of these information in the analysis. A further analysis of detailed drug usage information with outcomes are needed.

2 The authors disclosed that the cohort collected 372,000 aliquots of blood samples (plasma, buffy coat, and serum) from pregnant women during early pregnancy, 79,000 aliquots of urine samples, 30,000 aliquots of cord blood samples. This was a wonderful work but the authors need to describe more clearly about the biobank. Such as, Has the cohort collected biospecimen from all pregnant 112,986 women? How many pregnant women donated each kind of biospecimen in each visit? How many pregnant women donated complete sample across pregnancy? Is there any blank samples to determine potential sample contamination? How did the researchers transport the biospecimen across cities? What is the location and support strategy of the biobank? etc.

3 A comparison of basic characteristics between the initial cohort and finally included population is essential for this paper. Besides, A comparison between the DEBC and pregnant population of whole china is very helpful to stress the representativeness of this cohort. At least, A comparison between the DEBC and all pregnant population from the study hospitals during the study period is necessary. Also, the reviewer encourages the authors to add the comparison of the incidence rates of adverse outcomes in this population with other population in China and even the world.

4 Statistical methods are far from satisfied. There might be potential confounders, as shown in the 3rd Table. The authors should fully consider the potential confounders when studying the associations between drug usage and outcomes. Chi-square tests are essential but far from enough. Additionally, multiple comparison may bias the results.

Minor concerns:

1 The data control is key step of this work. Please clarify how did the author detect and deal with

outliers?

2 Revise the hidden number (“###”) in the 2nd Table.

3 Keep at least two decimal places when reporting rates, especially the drug exposure rates, considering some drug exposure rates are low.

4 Please clarify the meaning of %Complete in Tables.

5 Consider adding 95%CI of rates in tables.

6 Report the exact P values in the 3rd Table.

7 Clarify what kind of chi-square test () was used in the footnotes.

8 The forth table was split into two pages, which was not easy to review. Please reformat.

9 The tables and figures need titles and footnotes.

10 The Flow Chat. Drug Exposure Birth Cohort (DEBC) but not Drug Exposure China Cohort

11 The manuscript has several grammatical errors and this Reviewer suggest the authors thoroughly check the language with the help of a native English speaker or language service, in order to avoid these mistakes.

Reviewer #2 (Remarks to the Author):

In this manuscript, the authors describe the process of establishing a prospective longitudinal maternal drug exposure birth cohort in China, including detailed survey data, medical record data, and biological samples collected from the women at various time points in pregnancy. After almost 2.5 years of recruiting and follow-up, there is a total of 150,577 pregnant women included in the cohort. This is an incredible effort to generate a large cohort with detailed data on both medication exposure, maternal lifestyle and comorbidity data, as well as pregnancy and infant outcomes, creating an invaluable resource for research. Robust methodology for data and biological sample collection, storage, and quality control was used, and described well. Additional strengths of this cohort include information on whether the pregnancy was planned, as well as collection of data on exposure to Chinese medicines for which, as the authors note, there is limited information about their safety for use in pregnancy. The main limitation of the manuscript in its current version is the lack of information about the generalizability of this cohort of women to the general pregnant population in China, so I pose the following questions for the authors to address:

-how representative are the women in the cohort to the general pregnant population in the 15 provinces where the cohort was established and in China overall, with regards to medication use, age, lifestyle factors, education, comorbidities, etc? Do you have a strong selection bias, and how would this influence (bias) future research studies using the cohort?

-how many women receive antenatal care and deliver at the 49 collaborating medical institutions per year? Do these institutions care for a selected population which may limit the generalizability of the cohort (for example, they are high risk pregnancy centres?)

-recruiting 150,577 women in to the cohort is an impressive feat, however, are these numbers lower than expected considering the population size in the 15 provinces? What were the challenges and

limitations to recruiting even more women?

-please include information on 1) the number of women approached for inclusion into the study during recruitment, 2) the number of women who did not provide informed consent, and whether their characteristics differ from the women who did choose to participate (if known). This information could be included in the Figure 1 flowchart, for example.

-Additionally, include information on the number of women lost-to-follow up during the study period, and whether their characteristics differ from the women who completed all steps in the data collection. If a woman was lost-to-follow-up during the 3rd trimester, do you include her 1st trimester information in the cohort (for example, reporting the rates of early pregnancy exposure to medication)?

-In Flowchart Figure 1, 20,909 pregnant women were excluded due to data without quality audit. Please explain what this means. Additionally, do the characteristics of these women differ from the ones who were included in the final cohort?

Additional minor revision suggestions:

-In Figures 2A and 2B, the legend is unclear. Does 'human aliquots' mean the number of pregnant women from whom samples have been collected?

-Lines 118-125 describe the number of aliquots of the biological samples. More interesting to the reader would be the number of pregnant women these samples have been collected from, including the proportion of women that did not provide biological samples. In Figure 2A, it seems that fewer women provide samples over the trimesters of pregnancy – are these women lost-to-follow-up complete in the study, or did some of the provide responses to the survey and just not biospecimens?

-In the Figure 1 flowchart, it says that multifetal pregnancies ("multiple pregnancy") are excluded from the cohort, however Table S2 shows that samples were collected from "gemellary pregnancies". Please clarify.

-To Tables 1-3, please add the total number of women at the top of the column

-Table 1 reports 'parental smoking', do you have the data to report this information separately for the mother and partner? And specifically, if the smoking occurred during first trimester?

-For Table 4, how were these drugs chosen? They are not the same as the drugs with highest exposure rates listed in Table 2. Did you instead choose the drugs with the highest outcomes rates?

-The outcome status in Table 4 was reported according to 1st trimester exposure. Some of these outcomes (e.g., preterm birth) may be a result of medication exposure in later pregnancy. You may want

to note this in the discussion of these results.

-Please provide a reference at the end of the sentence on line 57, (unless you are referring to your cohort results? If so, please make this clear)

-Please provide references at the end of the sentence on line 295

-On line 231, it is stated that information on nutritional supplements such as folic acid were not collected. However, in Table 1 contains information on folic acid supplementation, please clarify.

- Is there a reference that you can include for readers to learn more about the definition of major malformation in the China Birth Defects Monitoring Program? How does that definition align with other malformation classification systems (e.g., EUROCAT)?

-As this is an invaluable resource for studying the safety of medication use in pregnant women, are there opportunities for investigators from other institutions to collaborate and conduct studies with data from this cohort? If so, I recommend adding information about this in the article.

-While generally well written, I recommend a thorough edit focusing on improving the English grammar as there are some incorrect verb tenses in the abstract and throughout the manuscript.

Reviewer #3 (Remarks to the Author):

This paper presents results from the maternal drug exposure birth cohort (DEBC) that was created and is being maintained in China. The current database has recruited 150,577 pregnant women. After excluding women without follow-up, quality audits, and multiple pregnancies information was analyzed on 112,986 pregnant women. There are several questions and concerns with the current version of the manuscript. Please see below for the main issues and suggestions.

Cited literature includes several review papers. Please cite original research when possible.

There seems to be a great deal of overlap between this study cohort and that included in a previously published study (China Teratology Birth Cohort (CTBC)). Please clarify between the two cohorts.

There is a lack of detail in the paper that is needed in order to assess the study outcomes. For instance, what was considered significant in the data analyses? How were the statistical analyses carried out? Women were recruited from 49 centers. Were site differences in treatment and outcomes explored or considered? It might explain some of the results.

Overall it would be helpful if Figures and Tables were numbered within the merged pdf.

There is an inconsistency in the listing of exclusion criteria in the abstract and methods section of the

manuscript. The abstract excludes twin pregnancies, but it is not mentioned in the Methods section; however, twin pregnancy information is available in one of the tables.

This study states that part of the purpose of this cohort is to establish a biobank of samples. Details are given for sample collection; however, there is no explanation as to how the samples will be used and what will be measured. For instance, all samples appear to be collected and stored, but there is no description of the tests to be run or when they might be run. The supplemental tables are difficult to read as they are expanded across two pages. Supple Table 3 indicates that samples will be stable “for years”. What do the authors mean by stable and how is this determined? What is to be measured will determine the relevance and practicality of sample stability.

The top 20 drugs identified as the most frequently used in pregnancy are somewhat unusual. The authors discuss the use of several of the medications, but do not discuss all of them. For instance, why is there such frequent use of heparin? It is mentioned that this medication is commonly used in clinical practice; however, this may not be true in all countries and further explanation is warranted. Table 2 has two numbers that need corrected. It appears that the numbers are too large for the cell.

It is mentioned that several of the medications have a high incidence with the rates of adverse pregnancy. Can the authors elaborate on this in the discussion? There are some questions concerning Table 4. There needs to be more explanation for this table. The results say that incidence of birth defects are “7.1% higher than 4.5% in the unexposed group”. Where is the information for the unexposed group? The right portion of the table seems to be missing. Also, why are all of the “No” groups in the table listed as 2.5%?

Point-to-point response

Reviewer #1

1. The pregnant women usually intake drug because of some medical condition, resulting in a huge bias when researching the associations between drug and adverse outcomes. The medical condition, but not the drug, might be the reason of the pregnant complications and adverse birth outcomes. The drug usage might actually reduce the risks of adverse outcomes when treated the medical conditions. It is of great importance to clarify this bias. The authors should be more rigorous when draw conclusion. Additionally, the cohort collected he drug exposure time, duration, route, dose and reason in the questionnaire. These are key information but it seems that the authors did not take fully consideration of these information in the analysis. A further analysis of detailed drug usage information with outcomes are needed.

Response: Thank you very much for these critical comments. As you have pointed out, the medical condition, but not the drug, might be the reason of the pregnant complications and adverse birth outcomes. We have clarified this important issue according to your suggestion in the paragraph of limitations in Discussion section [limitation (ii)] (lines 395-402, p.14).

Additionally, we have added descriptive analyses of drug formulation, route, dosage, and reasons of drug use in Table 2. However, in order to do systematic research and draw more rigorous conclusions, further analyses (considering exposure time, duration, route, dose and reasons of drug use) and presentation of results about correlations between metabolite exposures and outcomes will be performed in future studies.

2. The authors disclosed that the cohort collected 372,000 aliquots of blood samples (plasma, buffy coat, and serum) from pregnant women during early pregnancy, 79,000 aliquots of urine samples, 30,000 aliquots of cord blood samples. This was a wonderful work but the authors need to describe more clearly about the biobank. Such as, Has the cohort collected biospecimen from all pregnant 112,986 women? How many pregnant women donated each kind of biospecimen in each visit? How

many pregnant women donated complete sample across pregnancy? Is there any blank samples to determine potential sample contamination? How did the researchers transport the biospecimen across cities? What is the location and support strategy of the biobank? etc.

Response: Thank you very much for the comments and constructive suggestions. It is worth noting that all initially recruited 15,0577 pregnant women in the DEBC were asked to donate peripheral blood and urine samples on the first investigation. Eventually, among initially recruited 150,577 participants, about 83%, 22%, and 4% of pregnant women provided blood samples in the first trimester, the second trimester, and the third trimester, respectively; about 43%, 4%, and 4% of pregnant women provided their urine samples in the first trimester, the second trimester, and the third trimester, respectively. This information has been added into the revised manuscript (lines 125-130, p. 5).

Because we considered that the first trimester was the relatively most important period for fetal growth and development, of which the mistakes would lead to birth defects, the maternal peripheral blood samples and their processed samples (plasma, serum and buffy coat) during the first trimester accounted for the majority of the biospecimens in the biobank, and fewer biospecimens were collected during the second trimester and the fewest during the third trimester. This information was added in Discussion section (lines 226-231, p.8).

Based on our original research design, no blank samples (tubes) were stored. Yet, because the collection and processing of samples were carried out in clinical laboratories, in which the current national clinical laboratory standards and regulations were strictly followed, we judged that the possibility of sample contamination was very small, but we could not absolutely exclude the possibility. This has been acknowledged in the paragraph of limitations [limitation (iii)] in this revised manuscript (lines 402-407, p. 14).

All biospecimens were collected in ice boxes and processed in sub-centers within 2 hours after collection. Then each type of processed biospecimen from the same participant was processed into different vials which were immediately stored in -80°C refrigerators. The frozen biospecimens were placed in dry-ice boxes (temperature

ranging from -70°C to -78°C) which were transported across cities. This information was added in Quality Control section in Methods, as shown in lines 571-576, p. 19-20.

All processed peripheral blood samples and urine samples were finally stored in -80°C refrigerators, and biospecimens from placenta, muscular tissues and pathological tissues were stored in liquid nitrogen tanks at temperatures below -150°C in West China Second University Hospital. For final storage, multiple vials containing each type of processed biospecimens such as blood plasma, serum, and buffy coat from the same pregnant woman were not placed in the same box or refrigerator in principle, but were stored in different boxes and refrigerators to prevent unexpected problems caused by possible sample losses or breakdown of one refrigerator. This information has been added in Quality Control section in Methods, as shown in lines 578-586, p. 20.

3. A comparison of basic characteristics between the initial cohort and finally included population is essential for this paper. Besides, A comparison between the DEBC and pregnant population of whole China is very helpful to stress the representativeness of this cohort. At least, A comparison between the DEBC and all pregnant population from the study hospitals during the study period is necessary. Also, the reviewer encourages the authors to add the comparison of the incidence rates of adverse outcomes in this population with other population in China and even the world.

Response: Thank you for your suggestion. We have already added the comparison of basic characteristics between databases with 150,577 pregnant women (initial cohort) and analytical database with 112,986 pregnant women (added the Supplementary Table 2). It should be noted that currently only basic characteristic data of Chinese women of childbearing age could be obtained, and the detailed data on other basic characteristics of the entire pregnant population in China have not been reported. Therefore, the comparisons of basic characteristics between the DEBC data and the entire pregnant population in China could not be made. Besides, the investigators of this study did not have access to data on all pregnant women at the surveyed hospitals for comparison.

However, we obtained baseline data on two variables (maternal age and education

level) of all pregnant women at three DEBC collaborative hospitals from national monitoring data (The authors' institution is responsible for national monitoring data from China's National Maternal Near-Miss Surveillance System (NMNMSS)). We compared the basic characteristics from the DEBC data with those of all pregnant women at the three hospitals, and found statistically significant differences in age distribution but not in education level (see in Table R1). This may be because the three hospitals receive referrals of critically ill pregnant women, who are more likely to be of advanced maternal age, from other collaborative hospitals.

Table R1 Comparison of baseline characteristics between pregnant women in the DEBC data and the overall pregnant population at these three study hospitals

Baseline characteristics	N(%)		Chi-square value	P-value
	DEBC population	All pregnant from the three study hospitals		
Maternal age			80.45	<.0001
<20	58(0.22)	280(0.52)	38.41	<.0001
20-24	2122(8.06)	4530(8.47)	3.79	0.0515
25-29	9979(37.92)	19926(37.26)	3.36	0.0668
30-34	10314(39.2)	20138(37.65)	3814.67	<.0001
>=35	3840(14.59)	8610(16.1)	30.33	<.0001
Maternal education level			0.7127	0.3985
Junior high school and below	2379(9.04)	4327(8.85)		
High school and above	23943(90.96)	44543(91.15)		

Meanwhile, the incidence rates of adverse birth outcomes observed in this study were comparable to those reported in other domestic Chinese investigations and East Asian studies. In particular, the stillbirth rate aligned with East Asian estimates, whereas preterm birth and low birth weight incidences were marginally lower than Chinese surveillance statistics and research derived from East Asian data. The information has been added into this revised manuscript (lines 255-265, p. 9).

4. Statistical methods are far from satisfied. There might be potential confounders, as shown in the 3rd Table. The authors should fully consider the potential confounders when studying the associations between drug usage and outcomes.

Chi-square tests are essential but far from enough. Additionally, multiple comparison may bias the results.

Response: Thank you for your comments and suggestion. Indeed, univariate analysis has limitations due to potential confounding factors when examining associations between medications and outcomes. This important limitation has been clarified in the revised manuscript (lines 395-402, p. 14). To making more rigorous conclusions, we will conduct future studies by considering potential confounders to enable more in-depth investigations and a series of analyses of each drug type, allowing more robust analyses. Therefore, this manuscript only included univariate analyses as an initial exploratory investigation. We expect to attract collaborations with peers in similar fields to jointly conduct more follow-up studies (lines 418-424, p14-15, in the revised manuscript).

Minor concerns:

1. The data control is key step of this work. Please clarify how did the author detect and deal with outliers?

Response: Thank you for your question. We have added the relevant content to this revised manuscript (lines 560-568, p. 19). Data quality control is a critical step in our research. We regularly exported and analyzed the data to identify potential outliers. Considering that most quantitative data collected in this study followed a normal distribution, outliers were defined based on the 3-sigma principle (if the absolute residual error of a measurement exceeded 3σ , it was considered an outlier). For identified outliers, we first cross-checked against the hospital information system and national monitoring systems. For data that could not be verified, telephone follow-ups were conducted for confirmation and correction. For the small portion of outliers that simply could not be rectified through the above steps, given our large sample size and the rarity of such outliers, we treated them as missing data.

2. Revise the hidden number (“###”) in the 2nd Table.

Response: Thank you for your careful work. The number format might have changed during the generation of PDF files. We have revised the hidden number in the revised Table 2.

3. Keep at least two decimal places when reporting rates, especially the drug exposure rates, considering some drug exposure rates are low.

Response: Thank you for your constructive comment. We have made the revisions according to your suggestion in Tables 1, 2, 3, and Supplementary tables.

4. Please clarify the meaning of %Complete in Tables.

Response: % Complete represented the ratio in the group. We have replaced it with % in the revised manuscript in order to avoid ambiguity in Tables 1, 3, and Supplementary tables.

5. Consider adding 95%CI of rates in tables.

Response: Thank you for your suggestion. We have added 95% CI in Table 2 and Figure 3.

6. Report the exact P values in the 3rd Table.

Response: Thanks for your suggestion. We have added the exact *P* values that were available from statistical software in Table 3.

7. Clarify what kind of chi-square test () was used in the footnotes.

Response: Thanks. Pearson's chi-square test was used in the paper. As you recommended, we have explained this in the Methods section (line601, p. 21).

8. The forth table was split into two pages, which was not easy to review. Please reformat.

Response: Thank you for your suggestion. We have reformatted Table 4.

9. The tables and figures need titles and footnotes.

Response: We are very sorry for our negligence of the footnotes, and we have added the titles and footnotes in the figure legend in main text (lines804-853, p. 26-30).

10. The Flow Chat. Drug Exposure Birth Cohort (DEBC) but not Drug Exposure China Cohort.

Response: We are very sorry for our incorrect writing and we have corrected it to Drug Exposure Birth Cohort in the flow chart of Fig.1.

11. The manuscript has several grammatical errors and this Reviewer suggest the authors thoroughly check the language with the help of a native English speaker or language service, in order to avoid these mistakes.

Response: Thank you for your suggestion. We have checked the language problems thoroughly and corrected the grammar errors with the help of a native English speaker.

Reviewer #2

1. How representative are the women in the cohort to the general pregnant population in the 15 provinces where the cohort was established and in China overall, with regards to medication use, age, lifestyle factors, education, comorbidities, etc? Do you have a strong selection bias, and how would this influence (bias) future research studies using the cohort?

Response: Thank you very much for the questions. The answers to these questions would make the manuscript more informative.

The sample size of this study was calculated according to cohort study requirement during study design (Supplementary method), and the actual sample size that we had achieved greatly exceeded the statistical requirement of this study.

While the current sample size satisfied this study's requirements, the analytic cohort of 112,986 was still relatively small compared to China's approximately 10 million annual deliveries. Since comprehensive national data on the characteristics of pregnant women are unavailable, it remains uncertain whether the present cohort accurately represents the overall pregnant population in China.

However, this is to date the largest cohort with available actual data on maternal drug exposure during pregnancy and pregnancy outcomes in China, although the present study has not covered most pregnant women across China. Studies covering the entire or most pregnant population in China are not feasible. The factors such as limited financial resources and manpower increased the difficulty of further expanding the sample size for this project. Significantly, this study included collection of a large amount of examination results during pregnancy, and collection and preservation of various biospecimens, and multiple time points questionnaire surveys covering a total

of 1551 variables. However, based on the population demographics, incidence rates of birth outcomes, and the analysis of required sample size, the results of this study potentially represented the relatively most accurate assessment of medication use among pregnant women in China.

The incidence rates of adverse birth outcomes observed in this study were comparable to those reported in other domestic Chinese investigations and East Asian studies. In particular, the stillbirth rate aligned with East Asian estimates, whereas preterm birth and low birth weight incidences were marginally lower than Chinese surveillance statistics and research derived from East Asian data. Therefore, we do not think that there was a strong selection bias in this cohort.

The above information has been added in Discussion of the revised manuscript (lines 237-254, lines 260-265, p. 8-9; lines 411-417 p. 14).

2. how many women receive antenatal care and deliver at the 49 collaborating medical institutions per year? Do these institutions care for a selected population which may limit the generalizability of the cohort (for example, they are high risk pregnancy centres?)

Response: Thank you for the questions. Out of these 49 hospitals, approximately 190,000 pregnant women come for antenatal care per year, with 95% of them receiving antenatal care throughout their entire pregnancy and delivering their babies in these hospitals. Among those 95% of pregnant women, around 30% meet the important criteria of "gestational age less than 14 weeks upon first prenatal visit " and "willing to cooperate with the research and sign the informed consent" as well as other inclusion and exclusion criteria. These medical institutions care for the general population of pregnant women, not a selected sub-population. This information has been added into the revised manuscript (lines 440-444, p. 15).

3. recruiting 150,577 women in to the cohort is an impressive feat, however, are these numbers lower than expected considering the population size in the 15

provinces? What were the challenges and limitations to recruiting even more women?

Response: Thank you for the questions. We have added the relevant information related to the questions in this revised manuscript.

The sample size of this study was calculated according to cohort study requirement during study design (Supplementary method), and the actual sample size that we had achieved greatly exceeded the statistical requirement of this study.

This is to date the largest cohort with available actual data on maternal drug exposure during pregnancy and pregnancy outcomes in China, although the present study has not covered most pregnant women across China. Studies covering the entire or most pregnant population in China are not feasible. The factors such as limited financial resources and manpower increased the difficulty of further expanding the sample size for this project. Significantly, this study included collection of a large amount of examination results during pregnancy, and collection and preservation of various biospecimens, and multiple time points questionnaire surveys covering a total of 1551 variables. However, based on the population demographics, incidence rates of birth outcomes, and the analysis of required sample size, the results of this study potentially represented the relatively most accurate assessment of medication use among pregnant women in China. This information has been added in Discussion (lines 237-254, p. 8-9).

4. please include information on 1) the number of women approached for inclusion into the study during recruitment, 2) the number of women who did not provide informed consent, and whether their characteristics differ from the women who did choose to participate (if known). This information could be included in the Figure 1 flowchart, for example.

Response: Thank you for your suggestion. In the 49 collaborating hospitals, approximately 190,000 pregnant women come for antenatal care per year, with 95% of them receiving antenatal care throughout their entire pregnancy and delivering their babies in these hospitals. Among those 95% of pregnant women, all of them were

provided informed consents. However, around 30% of them meet the important criteria of "gestational age less than 14 weeks upon first prenatal visit " and "willing to cooperate with the research and sign the informed consent" as well as other inclusion criteria. This information has been added into the revised manuscript (lines 440-444, p. 15).

If the pregnant women did not meet inclusion criteria, their demographic data were not recorded anymore. Therefore, we do not have the detailed data on the difference in basic characteristics between general pregnant women and the those who choose to participate. But we obtained baseline data on two variables of DEBC cohort from three collaborating hospitals versus all pregnant women at those hospitals from national monitoring data (The authors' institution is responsible for national monitoring data from China's National Maternal Near Miss Surveillance System (NMNMSS)). We compared the basic characteristics from the DEBC data with those of all pregnant women at these three hospitals, and found statistically significant differences in age distribution but not in education level (see in Table R1). This might be because the three hospitals receive referrals of critically ill pregnant women, who are more likely to be of advanced maternal age (≥ 35 years old), from other collaborative hospitals.

Table R1 Comparison of baseline characteristics between pregnant women in the DEBC data and the overall pregnant population at these three study hospitals

Baseline characteristics	N(%)		Chi-square value	P-value
	DEBC population	All pregnant from the three study hospitals		
Maternal age			80.45	<.0001
<20	58(0.22)	280(0.52)	38.41	<.0001
20-24	2122(8.06)	4530(8.47)	3.79	0.0515
25-29	9979(37.92)	19926(37.26)	3.36	0.0668
30-34	10314(39.2)	20138(37.65)	3814.67	<.0001
≥ 35	3840(14.59)	8610(16.1)	30.33	<.0001
Maternal education level			0.7127	0.3985
Junior high school and below	2379(9.04)	4327(8.85)		
High school and above	23943(90.96)	44543(91.15)		

5. Additionally, include information on the number of women lost-to-follow up during the study period, and whether their characteristics differ from the women who completed all steps in the data collection. If a woman was lost-to-follow-up during the 3rd trimester, do you include her 1st trimester information in the cohort (for example, reporting the rates of early pregnancy exposure to medication)?

Response: Thank you for your suggestion and question. The number of pregnant women lost to follow-up during the study period was 9440 (~ 6.27%) out of 150577 pregnant women recruited in this study. The data were added into the Figure 1.

Since we included 112,986 pregnant women out of initially recruited 150,577 pregnant women for the current analysis based the reasons further explained in methods, the representativeness of the current analytical cohort was a concern. Our analysis showed that there were no statistically significant differences in most basic characteristics between databases with 150,577 pregnant women and the current analytical database with 112,986 pregnant women (Supplementary Table 2). This information has been added into the revised manuscript (lines 255-260, p. 9).

If a woman was lost to follow-up during the 3rd trimester, the pregnancy outcome was not available. Therefore, her data was judged incomplete. The women were then excluded from this analytic cohort. This information has been added in the revised manuscript (lines 546-549, p. 19).

Additionally, we also compared the basic characteristics such as age and education of 112,986 cases included in the analysis of this study and 9440 cases in the lost to follow-up group, and they did not show statistical differences (see in Table R2).

Table R2 Basic characteristics of the cases included in this study and those lost to follow-up

Baseline characteristics	Cases included		Cases lost to		Chi ²	P
	in the study		follow up			
	N	%	N	%		
Maternal age					2.014	0.733
<20	1116	1.01	94	1.02		
20-24	15308	13.82	1319	14.25		

25-29	44936	40.58	3700	39.97		
30-34	36476	32.94	3068	33.14		
>=35	12904	11.65	1076	11.62		
Maternal education level					0.294	0.909
Primary school and below	1353	1.20	116	1.23		
Junior high school	15405	13.67	1300	13.81		
High school	20554	18.23	1704	18.11		
College	68676	60.93	5730	60.89		
Postgraduate and above	6730	5.97	561	5.96		
Fertilization way					1.047	0.306
Natural conception	106597	94.68	8919	94.92		
Assisted reproduction	5993	5.32	477	5.08		
Plan for pregnancy					2.81	0.094
Yes	66690	59.21	5484	58.33		
No	45938	40.79	3918	41.67		

6. In Flowchart Figure 1, 20,909 pregnant women were excluded due to data without quality audit. Please explain what this means. Additionally, do the characteristics of these women differ from the ones who were included in the final cohort?

Response: Thank you for your suggestion and question. Perhaps we did not express ourselves clearly enough in the original manuscript. More detailed description about the exclusion criteria have been added in the revised manuscript (lines 549-551, p. 19) data failed to pass quality audit meant that the important data such as medication, pregnancy complications, neonatal delivery, etc. had not yet been verified in the hospital information system.

We compared the basic characteristics such as age and education of the 112,986 cases included in the analysis of this study with the 20,909 cases excluded in the current study because of the unfinished verification in quality control. The results showed no statistical differences (Table R3).

Table R3 Basic characteristics of the cases included in this study and those without quality control

Baseline characteristics	Cases included in the study	Cases without quality control	Chi ²	P
-----------------------------	-------------------------------	-------------------------	----------

	N	%	N	%		
Maternal age					1.193	0.879
<20	1116	1.01	207	1.01		
20-24	15308	13.82	2864	13.98		
25-29	44936	40.58	8262	40.32		
30-34	36476	32.94	6731	32.85		
>=35	12904	11.65	2428	11.85		
Maternal education level					4.145	0.387
Primary school and below	1353	1.20	279	1.34		
Junior high school	15405	13.67	2810	13.47		
High school	20554	18.23	3827	18.35		
College	68676	60.93	12662	60.71		
Postgraduate and above	6730	5.97	1277	6.12		
Fertilization way					0.555	0.456
Natural conception	106597	94.68	19756	94.80		
Assisted reproduction	5993	5.32	1083	5.20		
Plan for pregnancy					0.804	0.370
Yes	66690	59.21	12273	58.88		
No	45938	40.79	8571	41.12		

Additional minor revision suggestions:

1. In Figures 2A and 2B, the legend is unclear. Does ‘human aliquots’ mean the number of pregnant women from whom samples have been collected?

Response: Thank you for the question. Yes, the meaning of “Human aliquots” was not clear. We revised the term to processed biological samples in lines 123, 134, p. 5, and Figure 2, in the revised manuscript. In general, a processed blood sample such as plasma was further divided into four equal parts and stored in four tubes. In some special circumstances, plasma was divided into 2 or 3 tubes, such as blood collection volume of less than 4ml. The tube number was the actual number of tubes of which biological samples were stored.

2. Lines 118-125 describe the number of aliquots of the biological samples. More interesting to the reader would be the number of pregnant women these samples have been collected from, including the proportion of women that did not provide biological samples. In Figure 2A, it seems that fewer women provide samples over

the trimesters of pregnancy – are these women lost-to-follow-up complete in the study, or did some of the provide responses to the survey and just not biospecimens?

Response: Considering the actual situation in collaborating medical institutions and compliance of pregnant women, all recruited subjects were asked to provide biospecimens in the first trimester, but only a part of pregnant women from several specific medical institutions (such as Guangxi, Fujian and West China Second University Hospital) were asked to provide biospecimens in the middle and late trimester. Of the total number of 150577 pregnant women, 17% of pregnant women just provided the questionnaire information and did not provide their biological samples, 83% of pregnant women provided blood samples in the 1st trimester, 22% and 4% of pregnant women provided blood samples in the middle and the last trimester, respectively. For the urine samples, 43% of pregnant women provided their samples in the first trimester, and 4% provided the samples in the middle and the last trimester. This information was added into the revised manuscript (lines 126-130, p.5).

Further discussion about the data was added in the revised manuscript (lines 226-236, p.8) as follows: Because we considered that the first trimester was the relatively most important period for fetal growth and development, of which the mistakes would lead to birth defects, the maternal peripheral blood samples and their processed samples (plasma, serum and buffy coat) during the first trimester accounted for the majority of the biospecimens in the biobank, and fewer biospecimens were collected during the second trimester and the fewest during the third trimester. In the DEBC, 83%, 22%, and 4% of pregnant women provided blood samples in the first trimester, the second trimester and the third trimester, respectively. Totally, 4% of pregnant women provided their blood samples across the three trimesters of pregnancy. Regarding the urine samples, 43% of pregnant women provided their samples in the first trimester, and 4% in the second and third trimester.

3. In the Figure 1 flowchart, it says that multifetal pregnancies (“multiple pregnancy”) are excluded from the cohort, however Table S2 shows that samples were collected from “gemellary pregnancies”. Please clarify.

Response: 150577 pregnant women were initially recruited in the birth cohort. About

83% of initially recruited pregnant women donated their biospecimens voluntarily, and the sample collection followed the SOP (standard operating procedures) in the specimen collection, processing and storage. Therefore, the biological samples of pregnant women with multiple pregnancy including twins or triplets were also collected and stored in the biobank for future use (revised Supplementary Table 3). Multiple pregnancy was only excluded from this specific study on the relationship between drug exposure and adverse pregnancy outcomes (Fig. 1), because the unit of analysis was one fetus, and including multiple pregnancies would duplicate the calculation of a pregnant woman's exposure to a certain drug during pregnancy, which might introduce bias and affect the validity of the results. This information has been added into the revised manuscript (lines 555-559, p.19).

4. To Tables 1-3, please add the total number of women at the top of the column.

Response: Thank you for your suggestion. We have added the total number of women at the top of the column in Table 1, Table 2 and Table 3.

5. Table 1 reports ‘parental smoking’, do you have the data to report this information separately for the mother and partner? And specifically, if the smoking occurred during first trimester?

Response: Yes, “parental smoking” reported in Table 1 was defined as the occurring during the first trimester. We reported this information separately for the mother and partner, and the data have been added in Table 1 of the revised manuscript.

6. For Table 4, how were these drugs chosen? They are not the same as the drugs with highest exposure rates listed in Table 2. Did you instead choose the drugs with the highest outcomes rates?

Response: Thank you very much for pointing out this problem. Actually, the drugs in Tables 2 and 4 were identical, both being selected for the 24 with the highest exposure rates. The misunderstanding was caused by the fact that we translated the same individual drugs into different English names, and arranged them in different orders. We have re-ordered the drugs in Table 4 and re-checked the drug names, such as correcting the “Thyroid hormones” to “Thyroxine”, and the “Metformin” to “Dimethyldiguanide”.

7. The outcome status in Table 4 was reported according to 1st trimester exposure. Some of these outcomes (e.g., preterm birth) may be a result of medication exposure in later pregnancy. You may want to note this in the discussion of these results.

Response: Thank you very much for the suggestion. Yes, we agree that some of these outcomes may be a result of medication exposure in later pregnancy. We have added this in the paragraph of limitations [limitation (iv)] in the Discussion section, see lines 407-411, p. 14.

8. Please provide a reference at the end of the sentence on line 57, (unless you are referring to your cohort results? If so, please make this clear

Response: Thank you for the suggestion. Original line 57 was referring to our cohort results. We revised the sentence to make it more clear in revised line 62-63, p. 3.

9. Please provide references at the end of the sentence on line 295.

Response: Although proprietary Chinese medicines are widely used, there has been basically no systematic studies on pregnant women, except sporadic cases, which greatly weakened the evidence. We have added two references (35, 36) at the end of the sentence on line 338, p. 12, in this revised manuscript.

10. On line 231, it is stated that information on nutritional supplements such as folic acid were not collected. However, in Table 1 contains information on folic acid supplementation, please clarify.

Response: Thank you for your detailed work. In the present study, we collected the information of nutritional supplements during pregnancy in a separate part, such as folic acid, calcium and iron, and we used folic acid supplementation as a covariate to analyze the effects of drug exposure on pregnancy outcomes, etc.. Nutritional supplements were not classified as drugs according to Social Security Drug Classification and Codes (LD/T90-2012) in China. In order to better understand the scope of medication during pregnancy, we have supplemented this explanation in lines 589-592, p. 20, in the revised version.

11. Is there a reference that you can include for readers to learn more about the definition of major malformation in the China Birth Defects Monitoring Program?

How does that definition align with other malformation classification systems (e.g., EUROCAT)?

Response: Thank you for your question. Chinese Birth Defects Monitoring Network (CBDMN) primarily uses ICD10 for disease definitions and classifications, which is similar to the EUROCAT classification system for congenital anomalies, as exemplified in the following Table R4 (considering the limitation of the length of this response, the descriptions/definitions of only the nervous system, ear, eye, congenital heart defects are listed). Relevant references (the definitions and clinical features of the major birth defect) have been added to the Supplementary Reference for readers, as stated in line 598 in the revised manuscript.

Table R4 The comparison of birth defect description/definition between EUROCAT and CBDMN

	EUROCAT	CBDMN
Anencephaly	Total or partial absence of brain tissue and the cranial vault. The face and eyes are present. (incompatible with life)	A congenital malformation characterized by the total or partial absence of the cranial vault, the covering skin, and the brain missing or reduced to small mass. Includes: craniorachischisis. Includes: infants with iniencephaly and other neural tube defects as encephalocele or open spina bifida, when associated with anencephaly. Excludes: acephaly, that is, absence of head observed in amorphous acardiac twins.
Spina bifida	Midline defect of the osseous spine usually affecting the posterior arches resulting in a herniation or exposure of the spinal cord and/or meninges	a family of congenital malformation defects in the closure of the spinal column characterized by herniation or exposure of the spinal cord and/or meninges through an incompletely closed spine.
Encephalocele	Cystic expansion of meninges and brain tissue outside the cranium. Covered by normal or atrophic skin	a congenital malformation characterized by herniation of the brain and/or meninges through a defect in the skull. Encephalocele is not counted when present with spina bifida
Microcephaly	A reduction in the size of the brain with a skull circumference less than three standard deviations below the mean for sex, age and ethnic origin. Definitions known to vary between clinicians and regions.	a congenitally small cranium, defined by an occipito-frontal circumference (OFC) 3 standard deviation below the age- and sex-appropriate distribution curves. [If using a different definition or cut-off point (e.g., 2 standard deviations), report but specify criteria]. Excludes: microcephaly associated with anencephaly or encephalocele
Holoprosencephaly	Absence of the first cranial (olfactory) nerve tract. There is a spectrum of	a congenital malformation of the brain, characterized by various degrees of incomplete lobation of the brain

	anomalies from a normal brain, except for the first cranial nerve tract, to a single ventricle (holoprosencephaly)	hemispheres. Olfactory nerve tract may be absent. Holoprosencephaly includes cyclopia, ethmocephaly, cebocephaly, and premaxillary agenesis.
Hydrocephaly	Dilatation of ventricular system with impaired circulation and absorption of the cerebrospinal fluid. The dilatation should not be due to primary atrophy of the brain, with or without enlargement of the skull.	a congenital malformation characterized by dilatation of the cerebral ventricles, not associated with a primary brain atrophy, with or without enlargement of the head, and diagnosed at birth. Not counted when present with encephalocele or spina bifida. Excludes: macrocephaly without dilatation of ventricular system, skull of macerated fetus, hydranencephaly, holoprosencephaly, and postnatally acquired hydrocephalus.
Anophthalmos/microphthalmos	Unilateral or bilateral absence of the eye tissue. Clinical diagnosis/ Small eye/eyes with smaller than normal axial length. Clinical diagnosis	apparently absent or small eyes. Some normal adnexal elements and eyelids are usually present. In microphthalmia, the corneal diameter is usually less than 10 mm. and the antero-posterior diameter of the globe is less than 20 mm.
Anotia/microtia	Absent pinna, with or without atresia of ear canal	a congenital malformation characterized by absent parts of the pinna (with or without atresia of the ear canal) commonly expressed in grades (I-IV) of which the extreme form (grade IV) is anotia, absence of pinna. Excludes: small, normally shaped ears, imperforate auditory meatus with a normal pinna, dysplastic and low set ears.
Transposition of great vessels	Total separation of circulation with the aorta arising from the right ventricle and the pulmonary artery from the left ventricle	a cardiac defect where the aorta exits from the right ventricle and the pulmonary artery from the left ventricle, with or without other cardiac defects. Includes: double outlet ventricle so-called corrected transposition.
Tetralogy of Fallot	VSD close to the aortic valves, infundibular and pulmonary valve stenosis and overriding aorta across the VSD	a condition characterized by ventricular septal defect, overriding aorta, infundibular pulmonary stenosis, and often right ventricular hypertrophy.
Hypoplastic left heart syndrome	The definition includes atresia or marked hypoplasia of aortic orifice or valve with hypoplasia of ascending aorta and defective development of left ventricle (with or without mitral valve stenosis/atresia).	a cardiac defect with a hypoplastic left ventricle, associated with aortic and/or mitral valve atresia, with or without other cardiac defect.
Coarctation of the aorta	Constriction in the region of aorta where the ductus joins aorta	an obstruction in the descending aorta, almost invariably at the insertion of the ductus arteriosus

12. As this is an invaluable resource for studying the safety of medication use in pregnant women, are there opportunities for investigators from other institutions

to collaborate and conduct studies with data from this cohort? If so, I recommend adding information about this in the article.

Response: Thank you very much for the suggestion. Of course, we are open to collaborating with institutions or researchers who are interested and have a strong background in relevant research. Currently, we are in the phase of ensuring data quality and conducting data cleaning. In the future, we plan to conduct more research on medication use during pregnancy. Through collaboration, we hope to gather more evidence on the safety of medication use during pregnancy from different professional perspectives. We have added information about this at the end of the “Discussion” section, see line 418-424, p. 14-15.

13. While generally well written, I recommend a thorough edit focusing on improving the English grammar as there are some incorrect verb tenses in the abstract and throughout the manuscript.

Response: Thank you for your suggestion. We have checked the language problems thoroughly and corrected the grammar errors with the help of a native English speaker.

Reviewer #3

1. Cited literature includes several review papers. Please cite original research when possible.

Response: Thank you very much for your detailed work. The relevant references have been changed to the original research, and the revision is highlighted in yellow in the revised version.

2. There seems to be a great deal of overlap between this study cohort and that included in a previously published study (China Teratology Birth Cohort (CTBC)). Please clarify between the two cohorts.

Response: Thank you for your suggestion. China Teratology Birth Cohort (CTBC) was a hospital-based open-ended prospective cohort study with the aim to assess the risk of birth defects and other adverse pregnancy outcomes associated with maternal

environmental and behavioral exposures during pregnancy, which was expected to recruit at least 300,000 participants. The maternal drug exposure birth cohort (DEBC) was a sub-cohort based on CTBC, aimed to explore the impact of maternal drug exposure on pregnancy outcomes during pregnancy, and to establish a biobank of high-quality and well-annotated human biospecimens for further research on the mechanism of fetal health and diseases. DEBC had recruited 150,577 pregnant women and established a biobank with about 580,000 aliquots of human biological samples. The introduction of the two cohorts has been added and highlighted in yellow in the revised version (see lines 198-207, p. 7).

3. There is a lack of detail in the paper that is needed in order to assess the study outcomes. For instance, what was considered significant in the data analyses? How were the statistical analyses carried out? Women were recruited from 49 centers. Were site differences in treatment and outcomes explored or considered? It might explain some of the results.

Response: We highly appreciate the reviewer's constructive comments. In this revised manuscript, we have incorporated the following additional details:

(1) **Statistical analysis:** $P < 0.05$ or $P < 0.01$ were considered statistically significant. All statistical analyses were conducted using STATA® (Version 12.1, StataCorp LLC, College Station, TX, USA) and SAS® software (Version 9.4, SAS Institute Inc., Cary, NC, USA). This information has been added in lines 602-605, p. 21.

(2) **Study design and methods:** In order to provide readers with more detailed contents about the study design, we added the definition and coding of birth defects in the Chinese Birth Defects Monitoring Network (CBDMN) (Supplementary Reference, stated in line 597-598, p. 20), and also described the process of calculating the sample size for the cohort study (Supplementary Method, lines 252-254, p. 9).

(3) **Concerning geographical heterogeneity:**

In forthcoming analyses, we will incorporate geographic factors and employ multilevel modeling (with site as the level 1 unit) combining internal exposure and external exposures to conduct a series of analyses.

4. Overall it would be helpful if Figures and Tables were numbered within the merged pdf.

Response: Thank you for your careful work. Figures and Tables have been numbered in the revised version, while the number of Figures were not presented in the merged pdf. To make it inconvenient for you to review, we have added a reduction version of these figures in the figure legend in main text (lines 804-853, p. 26-30).

5. There is an inconsistency in the listing of exclusion criteria in the abstract and methods section of the manuscript. The abstract excludes twin pregnancies, but it is not mentioned in the Methods section; however, twin pregnancy information is available in one of the tables.

Response: Thank you very much for your comments. As you mentioned, twin pregnancy information is available in Supplementary Table 3 which was about subjects and types of specimen collection. About 83% of 150,577 initially recruited pregnant women donated their biospecimens voluntarily, and the sample collection followed the SOP (standard operative procedures) of the specimen collection, processing and storage. Therefore, the biological samples of pregnant women with multiple pregnancies were also collected and stored in our biobank. However, when analyzing the association between drug exposure and adverse pregnancy outcomes, we excluded multiple gestations (the analysis database contained 112,986 samples), because the unit of analysis was one fetus, and including multiple gestations would duplicate the calculation of a pregnant woman's exposure to a certain drug during pregnancy, which may introduce bias and affect the validity of the results. This information has been added into Methods section in the revised manuscript (lines 555-559, p. 19).

6. This study states that part of the purpose of this cohort is to establish a biobank of samples. Details are given for sample collection; however, there is no explanation as to how the samples will be used and what will be measured. For instance, all samples appear to be collected and stored, but there is no description

of the tests to be run or when they might be run. The supplemental tables are difficult to read as they are expanded across two pages. Supple Table 3 indicates that samples will be stable “for years”. What do the authors mean by stable and how is this determined? What is to be measured will determine the relevance and practicality of sample stability.

Response: Thank you very much for your questions. These biological samples were collected to establish the biobank. The collaborators of this project can utilize these different samples according to their purposes of research, such as genome detection with buffy coat, the environmental exposure measurement with the plasma, serum or urine, and pathological study with those tissues. We have performed a nested case-control study on SNPs related to drug metabolic enzyme and environmental exposure in preterm birth cases, which will be published in the near future. Otherwise, we would also conduct some multi-omics studies in the future, such as drug metabolomics with the plasma and urine. We also welcome other interested researchers for cooperation to explore more interesting topics using these abundant biological samples. The collaboration invitation has been added in lines 421-424, p. 14-15.

The original supplemental tables had typesetting problems, which have been resolved in the revised version. Regarding the stability of the biological samples, (Supplemental Table 4), our biological samples were preserved in ultra-low temperature (-80°C) environments since collection, and the DNA and proteins in biospecimens can remain relatively stable in years, which is important for future scientific research (two references are given below). In addition, we will perform relevant quality control before the study to ensure the stability and reliability of the samples. The method of quality control depends on the research purposes and the molecular types of measurements.

1. Shabihkhani M, Lucey GM, Wei B, et al.. The procurement, storage, and quality assurance of frozen blood and tissue biospecimens in pathology, biorepository, and biobank settings. *Clin Biochem* 2014;47:258–266.

2. Udtha M, Flores R, Sanner J, Nomie K, Backes E, Wilbers L, Caldwell J. The protection and stabilization of whole blood at room temperature. *Biopreserv Biobank*. 2014 Oct;12(5):332-6. doi: 10.1089/bio.2014.0026. PMID: 25340942.

7. The top 20 drugs identified as the most frequently used in pregnancy are somewhat unusual. The authors discuss the use of several of the medications, but do not discuss all of them. For instance, why is there such frequent use of heparin? It is mentioned that this medication is commonly used in clinical practice; however, this may not be true in all countries and further explanation is warranted. Table 2 has two numbers that need corrected. It appears that the numbers are too large for the cell.

Response: Thank you for your suggestion. The expression “The drugs are commonly used in clinical practice” in our original manuscript was not appropriate. We would like to convey that the exposure rate of these drugs was relatively high in the population of pregnant women in China. According to our data, the exposure rates of these drugs ranked in the top 20. We have revised this sentence in the manuscript (lines 348-349, p.12). We did not discuss all drugs, but only the top three with the highest exposure rates (Progesterone, Dydrogesterone, and Fuzheng medicament), as well as Clotrimazole, which we were interested in at the beginning of our design. There were two reasons. Firstly, our focus of this article was to present our efforts to establish a cohort of pregnancy drug exposures in China and the preliminary description of the exposures. Secondly, as we have emphasized, the current statistics did not exclude more confounding factors, and the results should be viewed with caution (line 395-402, p14). This information has been added in the paragraph of limitations in Discussion section. We will later conduct a special study on different drug classes, where more detailed statistics and more in-depth discussion will be made.

Regarding maternal exposure to heparin, we have added some discussion in this revised manuscript, see lines 352-362, p. 12: Thrombophilia has been implicated in adverse obstetrical events such as miscarriage, recurrent miscarriage, intrauterine growth restriction, severe pre-eclampsia, and placental abruption. There was also reasonable evidence to suggest that some cases of miscarriage and recurrent miscarriage were associated with thrombosis of placental vessels and infarction³⁷. Heparin can exert anti-clotting effects by increasing the action of antithrombin, the natural anticoagulant.

According to Chinese experts consensus on prevention and treatment of spontaneous abortion with low molecular weight heparin (LMWH) (2018) and Chinese expert consensus on diagnosis and management of recurrent spontaneous abortion (2022), LMWH and low dose aspirin (LDA) are recommended in China to reduce the risk of adverse pregnancy outcomes such as miscarriage, recurrent miscarriage and venous thromboembolism.

Additionally, we have adjusted the format of Table 2 to ensure a clear display of the numbers.

8. It is mentioned that several of the medications have a high incidence with the rates of adverse pregnancy. Can the authors elaborate on this in the discussion? There are some questions concerning Table 4. There needs to be more explanation for this table. The results say that incidence of birth defects are “7.1% higher than 4.5% in the unexposed group”. Where is the information for the unexposed group? The right portion of the table seems to be missing. Also, why are all of the “No” groups in the table listed as 2.5%?

Response: Based on the data shown in Table 4, it is indeed evident that the proportions of adverse pregnancy outcomes were higher in the group of pregnant women who took certain medications during early pregnancy.

The “NO” group meant the unexposed group in Table 4, defined as cases in the queue that were not exposed to this drug during early pregnancy. Not all of the “NO” groups were 2.5%, there were also 2.6% and 2.7%. For those 2.5%, they appeared to be equal because of the one decimal of the percentages, but the exact numbers were actually different. For example, the “NO” group 2.5% for Clotrimazole was actually 2.5271%, while the 2.5% for Allylestrenol was 2.5491%. The reason these numbers were so close was that the proportion of the pregnant women who did not take any drug out of the group of 112,986 subjects included in the analysis was the largest (about 70%), and the number of these drug-exposed individually in Table 4 would be very small in relative terms, so the proportions of unexposed groups were similar in terms of percentage.

The original Table 4 was too long and the right part got moved to the next page. In

this revised version, we have made the necessary changes to the formatting of Table 4. We also revised the original “YES” group to “exposed group” and the “NO” group to “unexposed group” for ease of interpretation.

REVIEWER COMMENTS

Reviewer #1 (Remarks to the Author):

The manuscript has been greatly improved. The authors revised all questions and gave satisfied responses to the majority of the comments. However, the key question of how to clarify the adverse outcome was caused by the drugs but not the pregnant complications was not clear. The reviewer was not convinced by a limitation clarification of the potential bias in the discussion considering this bias is a key factor which may result in a fake association of drug usage and adverse outcomes. Is there any other method or additional analysis to exclude or reduce this bias?

Reviewer #2 (Remarks to the Author):

The authors have satisfactorily responded to my comments and questions. The revised manuscript is much improved.

Reviewer #3 (Remarks to the Author):

The authors have added information to the manuscript that has improved its quality.

Wording in the Introduction refers to preliminary data. As this data is not accepted in a peer reviewed journal it would be more appropriate for the Results section and is not appropriate for the Introduction unless there is a published reference that is missing. Wording: In China, more than 30% of pregnant women had drug exposure history during the first trimester according to the preliminary data from our cohort. Lines 61-63

Lines 440-444. Please provide exact numbers for the number of women approached and consented as well as the number of samples in the bank.

Please clarify the meaning of the results for aspirin-, chorionic gonadotrophin-, heparin- and prednisone-exposed groups. Do you mean to say that those medications also had reduced incidence of miscarriage/abortion? Text: Meanwhile, the incidence of miscarriage/abortion was significantly reduced in the progesterone- or dydrogesterone-exposed group. The same results were obtained from aspirin-, chorionic gonadotrophin-, heparin- and prednisone-exposed group and unexposed group. Lines 179-180

Point-to-point response

Reviewer #1

The manuscript has been greatly improved. The authors revised all questions and gave satisfied responds to the majority of the comments. However, the key question of how to clarify the adverse outcome was caused by the drugs but not the pregnant complications was not clear. The reviewer was not convinced by a limitation clarification of the potential bias in the discussion considering this bias is a key factor which may result in a fake association of drug usage and adverse outcomes. Is there any other method or additional analysis to exclude or reduce this bias?

Response: We appreciate your constructive comments. Since the completion of the initial draft of our manuscript, our research team has been actively cleaning the database of health conditions of pregnant women. We have just finished the health condition data cleaning for pregnant women during early pregnancy. In response to your valuable suggestions, we have re-analyzed our data with maternal age and health condition information incorporated as covariates. The adjusted disease factors included diabetes, threatened miscarriage, infertility, thyroid disorders, common cold and influenza, vaginitis, thrombosis or antiphospholipid syndrome, hypertension, hepatitis, and other inflammatory diseases. We used log-binomial multivariate regression to estimate adjusted relative risks (aRR) for the association between maternal medication in the first trimester and adverse pregnancy outcomes, and the analysis results can be found in the revised Table 4. Additionally, the descriptions of the abstract (lines 42-49), results (lines 173-194), discussion (lines 392-411, lines 427-429) and statistical methods (lines 624-631) have been updated and highlighted in their respective sections.

Reviewer #2 (Remarks to the Author):

The authors have satisfactorily responded to my comments and questions. The revised manuscript is much improved.

Response: Thank you very much for your approval of our revised manuscript.

Reviewer #3 (Remarks to the Author):

1. The authors have added information to the manuscript that has improved its quality. Wording in the Introduction refers to preliminary data. As this data is not accepted in a peer reviewed journal it would be more appropriate for the Results section and is not appropriate for the Introduction unless there is a published reference that is missing. Wording: In China, more than 30% of pregnant women had drug exposure history during the first trimester according to the preliminary data from our cohort. Lines 61-63

Response: Thank you for your suggestion. This information has been described in the Results section (lines 152-153), therefore we have removed this sentence from the Introduction section.

2. Lines 440-444. Please provide exact numbers for the number of women approached and consented as well as the number of samples in the bank.

Response: Thank you for your thorough comment. The exact numbers for the number of women approached and consented as well as the number of samples in the bank have been added and highlighted in yellow in the revised version (lines 472-474).

3. Please clarify the meaning of the results for aspirin-, chorionic gonadotrophin-, heparin- and prednisone-exposed groups. Do you mean to say that those medications also had reduced incidence of miscarriage/abortion? Text: Meanwhile, the incidence of miscarriage/abortion was significantly reduced in the progesterone- or dydrogesterone-exposed group. The same results were obtained from aspirin-, chorionic gonadotrophin-, heparin- and prednisone-exposed group and unexposed group. Lines 179-180

Response: Thank you very much for your meticulous review. In this revised version, we used log-binomial multivariate regression to estimate adjusted relative risks (aRRs) for the association between maternal medication in the first trimester and adverse pregnancy outcomes. The adjusted disease factors included diabetes, threatened miscarriage, infertility, thyroid disorders, common cold and influenza, vaginitis, thrombosis or antiphospholipid syndrome, hypertension, hepatitis, and other inflammatory diseases. Following your suggestion, we have revised the description of

drug exposure and associated adverse pregnancy outcomes in a concise way, based on the updated statistical results after adjusting for maternal age and diseases (lines 173-194).

4. The discussion does a good job of reporting on drug exposure in other countries, but does not compare that to what is shown in patients in China. This can be useful as the medications shown in this paper as the most frequent are unusual compared to other studies. Is this a reflection of the type of patients treated in these hospitals or the population itself? The authors state that these were general hospitals.

Response: Thank you very much for your comments and questions. Currently, no other cohort studies on medication exposure during pregnancy in the Chinese population have been found. Only one study on medication use during pregnancy, based on data from the China Health Insurance Association (CHIRA) database, has been identified. In that study, the data about medication prescriptions in only 7,946 sampled pregnant women (2,896 pregnant women in the first trimester) was included in the analysis. Furthermore, the study did not investigate self-medication by pregnant women outside of hospitals. Although the comparability is limited, we added the medication exposure information in pregnant women from CHIRA study in our discussion to provide readers with a more comprehensive understanding (lines 284-291).

The study population in this study were pregnant women who received antenatal care and treatment in 49 hospitals (39 obstetrics, gynecology and pediatrics hospitals, and 10 general hospitals). In this study, approximately 95% of pregnant women received antenatal care throughout their entire pregnancy and delivered their babies. Pregnant women included in this study met the criteria of "gestational age not exceeding 14 weeks at the first antenatal visit" and "willing to participate in the study and sign the informed consent form." This meant that critically ill pregnant women transferred to these hospitals midway were not included in the study population. Therefore, the study population consisted of general pregnant women rather than a specific subgroup. This information has been described in the manuscript (lines 468-472).

REVIEWERS' COMMENTS

Reviewer #1 (Remarks to the Author):

The authors have addressed all the questions, and there are currently no further queries from the reviewer.

Reviewer #3 (Remarks to the Author):

The authors have adjusted the text to address previous concerns.